# LEARNING AND CONTROLLING THE SOURCE-FILTER REPRESENTATION OF SPEECH WITH A VARIATIONAL AUTOENCODER

## ABSTRACT

Understanding and controlling latent representations in deep generative models is a challenging yet important problem for analyzing, transforming and generating various types of data. In speech processing, inspiring from the anatomical mechanisms of phonation, the source-filter model considers that speech signals are produced from a few independent and physically meaningful continuous latent factors, among which the fundamental frequency and the formants are of primary importance. In this work, we show that the source-filter model of speech production naturally arises in the latent space of a variational autoencoder (VAE) trained in an unsupervised fashion on a dataset of natural speech signals. Using only a few seconds of labeled speech signals generated with an artificial speech synthesizer, we experimentally demonstrate that the fundamental frequency and formant frequencies are encoded in orthogonal subspaces of the VAE latent space and we develop a weakly-supervised method to accurately and independently control these speech factors of variation within the learned latent subspaces. Without requiring additional information such as text or human-labeled data, we propose a deep generative model of speech spectrograms that is conditioned on the fundamental frequency and formant frequencies, and which is applied to the transformation of speech signals.

## 1 INTRODUCTION

High-dimensional data such as natural images or speech signals exhibit some form of regularity which prevents their dimensions from varying independently from each other. This suggests that there exists a latent representation of smaller dimension from which the high-dimensional observed data were generated. Discovering the hidden properties of complex data is the goal of representation learning, and deep latent-variable generative models have emerged as promising unsupervised approaches (Goodfellow et al., 2014; Kingma & Welling, 2014; Rezende et al., 2014; Chen et al., 2016; Higgins et al., 2017; Kim & Mnih, 2018; Chen et al., 2018). The variational autoencoder (VAE) (Kingma & Welling, 2014; Rezende et al., 2014), which is equipped with both a generative and inference model, can be used not only for data generation but also for analysis and transformation. As an explicit model of a probability density function (pdf), the VAE can also be used as a learned prior for solving inverse problems such as compressed sensing (Bora et al., 2017), speech enhancement (Bando et al., 2018; Leglaive et al., 2018), or source separation (Kameoka et al., 2019; Jayaram & Thickstun, 2020). Making sense of the latent representation learned by a VAE and controlling the underlying continuous factors of variation in the data are important challenges to build more expressive and interpretable generative models and probabilistic priors.

Previous works on representation learning with deep generative models, in particular VAEs, have mostly focused on images (Higgins et al., 2017; Kim & Mnih, 2018; Chen et al., 2018; Locatello et al., 2019; 2020). Yet, it is not always easy to define the ground-truth latent factors of variation involved in the generation of natural images. For speech data, the latent factors of variation can be directly related to the anatomical mechanisms of speech production. This makes speech data interesting for investigating the disentangled representation learning capabilities of VAEs, complementary to studies dealing with images. A key concept for characterizing the structure of speech signals is deduced from the source-filter model proposed by Fant (1970). This model, described in

more detail in Section 2.2, implies that a speech signal is mainly characterized by a few continuous latent factors of variation corresponding to the vibration of the vocal folds (i.e., the source), which defines the fundamental frequency, and the resonances of the vocal tract (i.e., the filter), which define the formants. The source-filter model is at the core of various fundamental speech processing techniques such as cepstral representations and linear predictive coding (LPC) (Rabiner & Schafer, 2010). Valin & Skoglund (2019); Wang et al. (2019) and Juvela et al. (2019) have recently shown that the efficiency of neural speech vocoders can be largely improved by leveraging the source-filter model. Other works investigating the interaction between the source-filter model and neural networks include Lee et al. (2019) and Choi et al. (2021). All these studies illustrate the interest of combining deep learning techniques with more traditional signal processing models and algorithms. In this work, we interpret and control the latent space of a VAE from the perspective of the source-filter model of speech production, which can be beneficial for various applications in speech analysis, transformation, and synthesis.

We first train a VAE on a dataset of about 25 hours of unlabeled speech signals. Then, using only a few seconds of labeled speech signals generated with an artificial speech synthesizer, we propose a method to analyze and control the fundamental frequency and the formant frequencies in the latent representation of the previously trained VAE. Our contributions are the following: (i) We experimentally demonstrate that the fundamental frequency and the frequency of the first three formants are encoded in orthogonal subspaces of the VAE latent space. This shows that a vanilla VAE trained in an unsupervised fashion is able to learn a representation that is compliant with the source-filter model of speech production. (ii) We develop a weakly-supervised method to precisely and independently control the source-filter continuous latent factors of speech variation within the learned subspaces. We put in evidence the orthogonality of these subspaces, which allows us to perform speech transformations in a disentangled manner (i.e., modifying one of the factors does not affect the others). (iii) Without requiring additional information such as text or human-labeled data, we propose a deep generative model of speech spectrograms conditioned on the fundamental frequency and formant frequencies. To the best of our knowledge, this is the first study showing the link between the classical source-filter model of speech production and the representation learned in the latent space of a VAE. Thanks to this link, we propose a principled method to generate speech data controlled with interpretable trajectories (of e.g., fundamental frequency and formant frequencies).

## 2 BACKGROUND

### 2.1 VARIATIONAL AUTOENCODER

Generative modeling consists in learning a probabilistic model of an observable random variable $\mathbf{x} \in \mathcal{X} \subset \mathbb{R}^D$. Let $\mathcal{D} = \{\mathbf{x}_1, ..., \mathbf{x}_N \in \mathcal{X}\}$ be a dataset of $N = \#\mathcal{D}$ independent and identically distributed (i.i.d.) observations of $\mathbf{x}$. The empirical distribution of $\mathbf{x}$ is defined by $\hat{p}(\mathbf{x}) = \frac{1}{N} \sum_{\mathbf{x}_n \in \mathcal{D}} \delta(\mathbf{x} - \mathbf{x}_n)$, where $\delta$ is the Dirac delta function, which is null everywhere except in 0 where it takes the value 1.

The variational autoencoder (VAE) (Kingma & Welling, 2014; Rezende et al., 2014) attempts to approximate $\hat{p}(\mathbf{x})$ with a pdf $p_\theta(\mathbf{x})$ parametrized by $\theta$. High-dimensional data such as natural images or speech signals exhibit some form of regularity which prevents the $D$ dimensions of $\mathbf{x}$ from varying independently from each other. We can thus assume that there exists a latent variable $\mathbf{z} \in \mathbb{R}^L$, with $L \ll D$, from which the observed data were generated. Accordingly, the model distribution in the VAE is defined by marginalizing the joint distribution of the latent and observed data, $p_\theta(\mathbf{x}) = \int p_\theta(\mathbf{x}|\mathbf{z})p(\mathbf{z})d\mathbf{z}$.

In this work, the observed data vector $\mathbf{x} \in \mathbb{R}_+^D$ denotes the power spectrum of a short frame of speech signal (i.e., a column of the short-time Fourier transform (STFT) power spectrogram). Its entries are non negative and its dimension $D$ equals the number of frequency bins. We use the Itakura-Saito VAE (IS-VAE) (Bando et al., 2018; Leglaive et al., 2018; Girin et al., 2019) defined by

$$p(\mathbf{z}) = \mathcal{N}(\mathbf{z}; \mathbf{0}, \mathbf{I}), \qquad p_\theta(\mathbf{x}|\mathbf{z}) = \prod_{d=1}^{D} \text{Exp}\Big([\mathbf{x}]_d; [\mathbf{v}_\theta(\mathbf{z})]_d^{-1}\Big), \qquad (1)$$

where $\mathcal{N}$ and Exp denote the densities of the multivariate Gaussian and univariate exponential distributions, respectively, and $[\mathbf{v}]_d$ denotes the $d$-th entry of $\mathbf{v}$. The inverse scale parameters of $p_\theta(\mathbf{x}|\mathbf{z})$ are provided by a neural network called the decoder, parametrized by $\theta$ and taking $\mathbf{z}$ as input.

The marginal likelihood $p_\theta(\mathbf{x})$ and the posterior distribution $p_\theta(\mathbf{z}|\mathbf{x})$ are intractable due to the non linearities of the decoder, so it is necessary to introduce an inference model $q_\phi(\mathbf{z}|\mathbf{x}) \approx p_\theta(\mathbf{z}|\mathbf{x})$, which in the VAE is usually defined by

$$q_\phi(\mathbf{z}|\mathbf{x}) = \mathcal{N}\left(\mathbf{z}; \boldsymbol{\mu}_\phi(\mathbf{x}), \mathrm{diag}\{\mathbf{v}_\phi(\mathbf{x})\}\right), \tag{2}$$

where the mean and variance parameters are provided by a neural network called the encoder network, parametrized by $\phi$ and taking $\mathbf{x}$ as input. Then, the VAE training consists in maximizing a lower-bound of $\ln p_\theta(\mathbf{x})$, called the evidence lower-bound (ELBO) and defined by $\mathcal{L}(\theta, \phi) = \mathbb{E}_{\hat{p}(\mathbf{x})}\left[\mathbb{E}_{q_\phi(\mathbf{z}|\mathbf{x})}[p_\theta(\mathbf{x}|\mathbf{z})] - D_{\mathrm{KL}}\left(q_\phi(\mathbf{z}|\mathbf{x}) \parallel p(\mathbf{z})\right)\right]$. During training, the generative and inference model parameters $\theta$ and $\phi$ are jointly estimated by maximizing the ELBO, using (variants of) stochastic gradient descent with the so-called reparameterization trick (Kingma & Welling, 2014; Rezende et al., 2014).

## 2.2 SOURCE-FILTER MODEL OF SPEECH PRODUCTION

The source-filter model of speech production (Fant, 1970) is at the basis of many speech processing systems. It considers that the production of speech results from the interaction of a source signal with a linear filter. In voiced speech, the source originates from the vibration of the vocal folds, which produces a quasi-periodic glottal sound wave whose fundamental frequency defines the *pitch*. In unvoiced speech, a noise source is produced by a turbulent airflow or an acoustic impulse. The source signal is modified by the vocal tract, which is assumed to act as a linear filter. The cavities of the vocal tract give rise to resonances, which are called the *formants* and are characterized by their frequency, amplitude and bandwidth. By moving the speech articulators such as the tongue, lips, and jaw, humans modify the shape of their vocal tract, which results in a change of the acoustic filter and the associated resonances. This is how the different elementary speech sounds called phonemes are produced to form syllables, words and sentences.

The power spectra and the spectral envelopes of two French vowels are displayed in Figure 1. The spectral envelopes show that the formant frequencies are different for the two vowels. In this example however, the harmonic structure of the spectra shows that the fundamental frequency is the same for the two vowels. Formant frequencies are important distinctive features of vowels. In a first approximation, they can be related to the opening of the mouth, the front/rear position of the tongue, and the rounding of the lips for the first, second, and third formant respectively. For voiced phonemes, humans are able to control the formants independently of the pitch (i.e., to change the filter independently of the source (Fant, 1970)) and of each other (MacDonald et al., 2011). The independence of the source and filter characteristics makes the speech signals an interesting material for representation learning methods, especially with deep generative latent-variable models.

In the present study, in addition to the pre-trained IS-VAE speech spectrogram model, we also assume the availability of an artificial speech synthesizer allowing for an accurate and independent control of the fundamental frequency and formants. In this work, we use Soundgen (Anikin, 2019), a parametric synthesizer based on the source-filter model of speech production. For a given speech sound, the voiced component of the source signal is generated by a sum of sine waves, the noise component by a filtered white noise, and both components are then summed and passed through a linear filter simulating the effect of the human vocal tract. Importantly, this synthesizer allows us to easily generate artificial speech data labeled with the fundamental frequency and formant frequency values.

## 3 ANALYZING AND CONTROLLING SOURCE-FILTER FACTORS OF SPEECH VARIATION IN THE VAE

In this section, from a VAE trained on natural speech signals and a few artificially-generated labeled speech signals, we propose (i) a method to learn latent subspaces encoding source-filter factors of speech variation, (ii) a simple strategy to measure the disentanglement of the learned representation, and (iii) a weakly-supervised approach to control the continuous factors of variation in the learned subspaces and generate corresponding speech signals.

Let $f_i$ denote the speech factor of variation (in Hz) corresponding to the fundamental frequency, for $i = 0$, and to the formant frequencies, for $i \in \{1, 2, ...\}$. Let $\mathcal{D}_i$ denote a dataset of artificially-generated speech vectors (more precisely short-term power spectra) synthesized by varying only $\tilde{f}_i$,

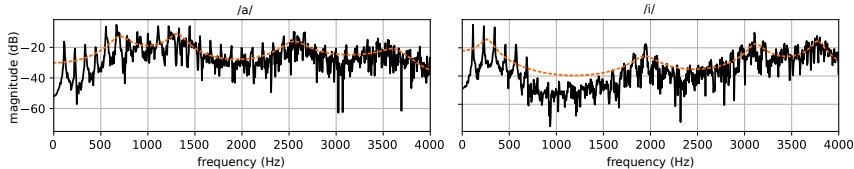

Figure 1: Power spectrum (solid black line) and spectral envelop (orange dashed line) for two vowels uttered by a male speaker.

all other factors $\{f_j, j \neq i\}$ being arbitrarily fixed. All examples in $\mathcal{D}_i$ are labeled with the factors of variation. It would be very difficult to build such a dataset from existing corpora of unlabeled natural speech. In contrast, it is an easy task using an artificial speech synthesizer such as Soundgen, which precisely takes the fundamental frequency and formant parameters as input, and outputs waveforms from which we extract power spectra.

### 3.1 LEARNING LATENT SUBSPACES ENCODING SOURCE-FILTER FACTORS OF VARIATION

Let $\hat{p}^{(i)}(\mathbf{x})$ denote the empirical distribution associated with $\mathcal{D}_i$, defined similarly as $\hat{p}(\mathbf{x})$. We also introduce the following marginal distribution over the latent vectors:

$$\hat{q}_\phi^{(i)}(\mathbf{z}) = \int q_\phi(\mathbf{z}|\mathbf{x})\hat{p}^{(i)}(\mathbf{x})d\mathbf{x} = \frac{1}{\#\mathcal{D}_i} \sum_{\mathbf{x}_n \in \mathcal{D}_i} q_\phi(\mathbf{z}|\mathbf{x}_n). \tag{3}$$

In the literature, this quantity is referred to as the aggregated posterior (Makhzani et al., 2016). However, $q_\phi(\mathbf{z}|\mathbf{x})$ is usually aggregated over the empirical distribution $\hat{p}(\mathbf{x})$ such that the aggregated posterior is expected to match with the prior $p(\mathbf{z})$ (Chen et al., 2018; Dai & Wipf, 2018). In contrast, in Equation (3) we aggregate over the "biased" data distribution $\hat{p}^{(i)}(\mathbf{x})$, where we know only one latent factor varies. This defines the explicit inductive bias (Locatello et al., 2019) that we exploit to learn the latent source-filter representation of speech in the VAE.

In the following of the paper, without loss of generality, we assume that, for each data vector in $\mathcal{D}_i$, the associated latent vector $\mathbf{z}$ has been centered by subtracting the mean vector

$$\boldsymbol{\mu}_\phi(\mathcal{D}_i) = \mathbb{E}_{\hat{q}_\phi^{(i)}(\mathbf{z})}[\mathbf{z}] = \frac{1}{\#\mathcal{D}_i} \sum_{\mathbf{x}_n \in \mathcal{D}_i} \boldsymbol{\mu}_\phi(\mathbf{x}_n). \tag{4}$$

Because only one factor varies in $\mathcal{D}_i$, we expect latent vectors drawn from the "biased" aggregated posterior in Equation (3) to live on a lower-dimensional manifold embedded in the original latent space $\mathbb{R}^L$. We assume this manifold to be a subspace characterized by its semi-orthogonal basis matrix $\mathbf{U}_i \in \mathbb{R}^{L \times M_i}$, $1 \leq M_i < L$. This matrix is computed by solving the following optimization problem:

$$\min_{\mathbf{U} \in \mathbb{R}^{L \times M_i}} \mathbb{E}_{\hat{q}_\phi^{(i)}(\mathbf{z})} \left[ \left\| \mathbf{z} - \mathbf{U}\mathbf{U}^\top \mathbf{z} \right\|_2^2 \right], \qquad s.t. \ \mathbf{U}^\top \mathbf{U} = \mathbf{I}. \tag{5}$$

The space spanned by the columns of $\mathbf{U}_i$ is a subspace of the original latent space $\mathbb{R}^L$ in which the latent vectors associated with the variation of the factor $f_i$ in $\mathcal{D}_i$ are expected to live. In Appendix A, we show that, similarly to principal component analysis (PCA) (Pearson, 1901), the solution to the optimization problem (5) is given by the $M_i$ eigenvectors corresponding to the $M_i$ largest eigenvalues of

$$\mathbf{S}_\phi(\mathcal{D}_i) = \frac{1}{\#\mathcal{D}_i} \sum_{\mathbf{x}_n \in \mathcal{D}_i} \left[ \boldsymbol{\mu}_\phi(\mathbf{x}_n)\boldsymbol{\mu}_\phi(\mathbf{x}_n)^\top + \text{diag}\{\mathbf{v}_\phi(\mathbf{x}_n)\} \right] - \boldsymbol{\mu}_\phi(\mathcal{D}_i)\boldsymbol{\mu}_\phi(\mathcal{D}_i)^\top. \tag{6}$$

The dimension $M_i$ of the subspace can be chosen such as to retain a certain percentage of the data variance in the latent space. Note that the only source of supervision used here is the knowledge that only the factor $f_i$ varies in the dataset $\mathcal{D}_i$.

### 3.2 DISENTANGLEMENT ANALYSIS OF THE LATENT REPRESENTATION

As defined by Higgins et al. (2018), a representation is disentangled if it is possible to learn orthogonal latent subspaces associated with each factor of variation, whether they are single- or multi-dimensional. The approach presented in the previous subsection exactly follows this definition and

offers a natural and straightforward way to objectively measure if the unsupervised VAE managed to learn a disentangled representation of the factors of variation under consideration. First, by simply looking at the eigenvalues associated with the columns of $\mathbf{U}_i \in \mathbb{R}^{L \times M_i}$, we can measure the amount of variance that is retained by the projection $\mathbf{U}_i \mathbf{U}_i^\top$. If a small number of components $M_i$ represents most of the variance, it indicates that only a few intrinsic dimensions of the latent space are dedicated to the factor of variation $f_i$ and varying this factor can be done by affine transformations. Second, if for two different factors of variation $f_i$ and $f_j$, with $i \neq j$, the columns of $\mathbf{U}_i$ are orthogonal to those of $\mathbf{U}_j$, this indicates that the two factors are encoded in orthogonal subspaces and therefore disentangled. It should however be verified experimentally that applying transformations by moving onto the subspace associated with $f_i$ generalizes to values of $\{f_j, j \neq i\}$ different than the ones used in $\mathcal{D}_i$.

### 3.3 CONTROLLING THE FACTORS OF VARIATION FOR SPEECH TRANSFORMATION AND GENERATION

So far, for each factor $f_i$, we have defined a methodology to learn a latent subspace $\mathbf{U}_i \in \mathbb{R}^{L \times M_i}$ that encodes its variations in the dataset $\mathcal{D}_i$, containing a few examples of speech data generated by an artificial synthesizer. Making now use of the labels in $\mathcal{D}_i$, we learn a regression model $\mathbf{g}_{\eta_i}$ : $\mathbb{R}_+ \mapsto \mathbb{R}^{M_i}$ from the factor $f_i$, whose value is denoted by $y \in \mathbb{R}_+$, to the data coordinates in the latent subspace defined by $\mathbf{U}_i$. The parameters $\eta_i$ are defined as the solution of the following optimization problem:

$$\min_\eta \left\{ \mathbb{E}_{\hat{q}_\phi^{(i)}(\mathbf{z},y)} \left[ \left\| \mathbf{g}_\eta(y) - \mathbf{U}_i^\top \mathbf{z} \right\|_2^2 \right] \overset{c}{=} \frac{1}{\#\mathcal{D}_i} \sum_{(\mathbf{x}_n,y_n) \in \mathcal{D}_i} \left\| \mathbf{g}_\eta(y_n) - \mathbf{U}_i^\top \left( \boldsymbol{\mu}_\phi(\mathbf{x}_n) - \boldsymbol{\mu}_\phi(\mathcal{D}_i) \right) \right\|_2^2 \right\}, \tag{7}$$

where $\hat{q}_\phi^{(i)}(\mathbf{z},y) = \int q_\phi(\mathbf{z}|\mathbf{x}) \hat{p}^{(i)}(\mathbf{x},y) d\mathbf{x}$, $\hat{p}^{(i)}(\mathbf{x},y)$ is the empirical distribution associated with $\mathcal{D}_i$, considering now both the speech data vector $\mathbf{x}$ and the label $y$, and $\overset{c}{=}$ denotes equality up to an additive constant w.r.t. $\eta$. The dataset $\mathcal{D}_i$ is very small with only a few hundreds examples, and as it is synthetic and labels are not provided by human annotators, the problem can be considered very weakly supervised. For simplicity and because it revealed efficient for this task, $\mathbf{g}_{\eta_i}$ is chosen as a piece-wise linear regression model learned independently for each output coordinate $m \in \{1, ..., M_i\}$. This choice is supported by the fact that the semi-orthogonal matrix $\mathbf{U}_i$ decorrelates the data (Bengio et al., 2013). Solving the optimization problem (7) then basically consists in solving a linear system of equations (Jekel & Venter, 2019).

We can now transform a speech spectrogram by analyzing it with the VAE encoder, then linearly moving in the learned subspaces using the above regression model, and finally resynthesizing it with the VAE decoder. Given a source latent vector $\mathbf{z}$ and a target value $y$ for the factor $f_i$, we apply the following affine transformation:

$$\tilde{\mathbf{z}} = \mathbf{z} - \mathbf{U}_i \mathbf{U}_i^\top \mathbf{z} + \mathbf{U}_i \mathbf{g}_{\eta_i}(y). \tag{8}$$

This transformation consists in (i) subtracting the projection of $\mathbf{z}$ onto the subspace associated with the factor of variation $f_i$; and (ii) adding the target component provided by the regression model $\mathbf{g}_{\eta_i}$ mapped from the learned subspace to the original latent space by the matrix $\mathbf{U}_i$. This operation allows us to move only in the latent subspace associated with the factor $f_i$. If this subspace is orthogonal to the latent subspaces associated with the other factors $\{f_j, j \neq i\}$, the latter should remain the same between $\mathbf{z}$ and $\tilde{\mathbf{z}}$, only $f_i$ should be modified. This process can be straightforwardly generalized to multiple factors, by subtracting and adding terms corresponding to each one of them.

Finally, as the prior $p(\mathbf{z})$ and inference model $q_\phi(\mathbf{z}|\mathbf{x})$ are Gaussian (see Equations (1) and (2)), the transformation in Equation (8) has the following probabilistic formulation (using $\mathbf{U}_i^\top \mathbf{U}_i = \mathbf{I}$):

$$p(\tilde{\mathbf{z}}; f_i = y) = \mathcal{N}\left( \tilde{\mathbf{z}}; \mathbf{U}_i g_{\eta_i}(y), \mathbf{I} - \mathbf{U}_i \mathbf{U}_i^\top \right) \tag{9}$$

$$q_\phi(\tilde{\mathbf{z}}|\mathbf{x}; f_i = y) = \mathcal{N}\left( \tilde{\mathbf{z}}; \mathbf{U}_i g_{\eta_i}(y) + (\mathbf{I} - \mathbf{U}_i \mathbf{U}_i^\top) \boldsymbol{\mu}_\phi(\mathbf{x}), (\mathbf{I} - \mathbf{U}_i \mathbf{U}_i^\top) \operatorname{diag}\{\mathbf{v}_\phi(\mathbf{x})\} \right). \tag{10}$$

The prior in Equation (9) is now conditioned on the factor $f_i$ and can be used to generate speech data given input trajectories of fundamental frequency and formant frequencies. As we assumed centered latent data, the mean vector $\boldsymbol{\mu}_\phi(\mathcal{D}_i)$ defined in Equation (4) must be added to $\tilde{\mathbf{z}}$ before mapping this vector through the generative model $p_\theta(\mathbf{x}|\mathbf{z})$.

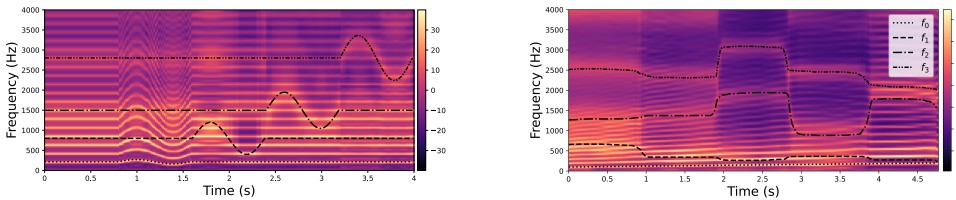

(a) Fundamental frequency and formant transforma- (b) Spectrogram generated from input trajectories of
tions of the vowel /a/ uttered by a female speaker. the fundamental frequency and formant frequencies.

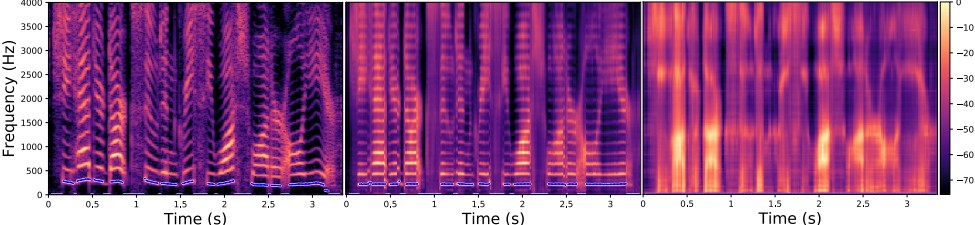

(c) Left: original spectrogram of a speech signal uttered by a female speaker; Middle: transformed spectrogram
where the fundamental frequency is set constant over time; Right: transformed spectrogram where the original
voiced speech signal (left) is converted into a whispered speech signal (i.e., the pitch is removed).

Figure 2: Qualitative example of modified and generated spectrograms with the proposed method.
The color bar indicates the power in dB.

## 4 EXPERIMENTS

This section presents qualitative and quantitative experimental results of the proposed method for
controlling the fundamental frequency and formant frequencies of speech signals with a VAE. The
VAE is trained on about 25 hours of multi-speaker speech data from the Wall Street Journal (WSJ0)
dataset (Garofolo et al., 1993a). The data space dimension is 513 and the latent space dimension is
16. For a given factor of variation, the corresponding latent subspace is learned (see Section 3.1)
using short trajectories of speech power spectra (corresponding to a few seconds of speech) gen-
erated with Soundgen (Anikin, 2019), all other factors being arbitrarily fixed. When solving the
optimization problem (5), the latent subspace dimension $M_i$ of each factor of variation is chosen
such that 80% of the data variance is retained. This leads $M_0 = 4$, $M_1 = 1$ and $M_2 = M_3 = 3$. The
regression models used to control the speech factors of variation in the latent space (see Section 3.3)
are learned on the same trajectories, but using the labels that correspond to the input control parame-
ters of Soundgen (i.e., fundamental frequency and formant frequencies values). More details on the
experimental set-up can be found in Appendix B. Given a generated or transformed spectrogram,
we use Waveglow (Prenger et al., 2019) to reconstruct the time-domain signal.

### 4.1 QUALITATIVE RESULTS

In Figure 2a, we illustrate the ability of the proposed method to modify the fundamental frequency
and formant frequencies in an accurate and independent manner. The spectrogram contains five
segments of equal length. The first segment corresponds to the original spectrogram of the steady
vowel /a/ uttered by a female speaker. In the following segments, we vary successively each indi-
vidual factor $f_i$, for $i = 0$ to 3, as indicated by the black lines in the figure. Variations of $f_0$ modify
the harmonic structure of the signal while keeping the formant structure unaltered. Variations of $f_i$,
$i \in \{1, 2, 3\}$, modify the formant frequencies (i.e., the vocal tract resonances, as indicated by the
color map) while keeping the fundamental frequency unaltered. Figure 2b represents a spectrogram
generated by using the conditional prior in equation (9) (generalized to conditioning on multiple fac-
tors). We can see that the characteristics of the generated speech spectrogram match well with the
input trajectories represented by the lines in the figure. In Figure 2c, from left to right we show the
original spectrogram of a speech signal uttered by a female speaker (left), and the transformed spec-
trograms where the fundamental frequency is set constant over time (middle) and where the pitch
has been removed (i.e., the original voiced speech signal is converted into a whispered speech signal)
(right). This last spectrogram is simply obtained by subtracting to **z** its projection onto the latent

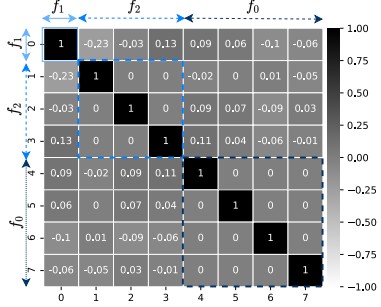

Figure 3: Correlation matrix of the learned latent subspaces basis vectors.

Table 1: Performance for pitch ($f_0$) and formant ($f_1, f_2$) transformations of English vowels.

| Factor | Method | NISQA (↑) | MOSnet (↑) | $\delta f_0$ (%, ↓) | $\delta f_1$ (%, ↓) | $\delta f_2$ (%, ↓) |
|--------|--------|-----------|------------|---------------------|---------------------|---------------------|
| $f_0$ | TD-PSOLA | $2.32 \pm 0.55$ | $2.60 \pm 0.36$ | $3.8 \pm 2.5$ | $6.3 \pm 2.8$ | $3.7 \pm 0.9$ |
| | WORLD | $2.49 \pm 0.60$ | $2.46 \pm 0.46$ | $4.5 \pm 0.6$ | $3.7 \pm 1.8$ | $2.3 \pm 0.7$ |
| | VAE baseline | $1.94 \pm 0.43$ | $2.44 \pm 0.28$ | $6.2 \pm 2.8$ | $10.4 \pm 2.4$ | $6.2 \pm 0.9$ |
| | Proposed | $2.08 \pm 0.48$ | $2.55 \pm 0.40$ | $0.8 \pm 0.2$ | $7.2 \pm 1.3$ | $3.6 \pm 1.2$ |
| $f_1$ | VAE baseline | $1.84 \pm 0.50$ | $2.40 \pm 0.10$ | $11.3 \pm 4.2$ | $15.1 \pm 3.5$ | $6.0 \pm 1.2$ |
| | Proposed | $1.85 \pm 0.40$ | $2.45 \pm 0.12$ | $6.0 \pm 1.6$ | $8.4 \pm 3.2$ | $5.7 \pm 0.4$ |
| $f_2$ | VAE baseline | $2.01 \pm 0.40$ | $2.40 \pm 0.08$ | $19.5 \pm 3.2$ | $10.7 \pm 0.5$ | $10.9 \pm 1.9$ |
| | Proposed | $2.03 \pm 0.43$ | $2.42 \pm 0.41$ | $8.5 \pm 1.1$ | $8.7 \pm 1.1$ | $6.2 \pm 1.5$ |

subspace corresponding to $f_0$ (i.e., by considering only the two first terms in the right-hand side of Equation (8)). It results in a spectrogram where the harmonic component is neutralized, while preserving the original formant structure. This is remarkable considering that the VAE was not trained on whispered speech signals, and it further confirms that the proposed method dissociates the source and the filter contributions in the VAE latent space. Other qualitative results can be found in Appendix C (visualization of trajectories in the learned latent subspaces and additional examples of generated and transformed speech spectrograms) and at `https://tinyurl.com/iclr2022` (including audio examples).

## 4.2 QUANTITATIVE RESULTS

Due to space limitations, we only provide in this section quantitative results related to the fundamental frequency ($f_0$) and the first two formant frequencies ($f_1$ and $f_2$). Additional results can be found in Appendix D, including the performance for modifications of the third formant frequency ($f_3$), the performance with a VAE trained on other datasets (e.g., a dataset of French speech signals), and the performance on the TIMIT dataset (Garofolo et al., 1993b), which is richer than the English vowel one in terms of phonemes but is not labeled with the fundamental and formant frequencies, making the evaluation less reliable.

### 4.2.1 ORTHOGONALITY OF THE LATENT SUBSPACES

Following the discussion in Section 3.2, we compute the dot product between all pairs of unit vectors in the matrices $\mathbf{U}_i \in \mathbb{R}^{L \times M_i}$, $i \in \{0, 1, 2\}$. Figure 3 shows that the resulting correlation matrix is mainly diagonal. Except for a correlation value of $-0.23$ across $f_1$ and the first component of $f_2$, all other values are below $0.13$ (in absolute value), confirming the orthogonality of the learned subspaces and thus the disentanglement of the learned source-filter representation of speech.

### 4.2.2 PITCH AND FORMANT TRANSFORMATIONS OF ENGLISH VOWELS

**Experimental set-up** In this experiment, we quantitatively evaluate the performance of the proposed method regarding the modification of the fundamental frequency and formant frequencies in speech signals (see Section 3.3). We use a corpus of 12 English vowels uttered by 50 male and 50 female speakers (Hillenbrand et al., 1995). This dataset is labeled with the fundamental and formant frequencies. We transform each signal in this dataset by varying one single factor $f_i$ at a time, for $i \in \{0, 1, 2\}$. The ranges of transformation for the fundamental frequency, first formant frequency, and second formant frequency are respectively $[100, 300]$ Hz, $[300, 900]$ Hz, and $[1100, 2700]$ Hz, with a step of 1 Hz, 10 Hz and 20 Hz respectively. For the modification of each factor $f_i$, we measure the performance regarding three aspects: First, in terms of *accuracy* by comparing the target value for the factor (see Equation (8)) and its estimation computed from the modified output speech signal. Second, in terms of *disentanglement*, by comparing the values of $f_j$ for $j \neq i$, before and after modification of the factor $f_i$. Third, in terms of speech *naturalness* of the transformed signal.

**Metrics** Accuracy and disentanglement are measured in terms of relative absolute error (in percent, the lower the better). For a given factor $f_i$, it is defined by $\delta f_i = 100\% \times |\hat{y} - y|/y$ where $y$ is

the target value of $f_i$ and $\hat{y}$ its estimation from the output transformed signal. Let us take the example of a modification of $f_0$: $\delta f_0$ measures the accuracy of the transformation on $f_0$ while $\delta f_1$ and $\delta f_2$ are used to assess if the other factors of variation $f_1$ and $f_2$ remained unchanged after modifying $f_0$. We use CREPE (Kim et al., 2018) to estimate the fundamental frequency and Parselmouth (Jadoul et al., 2018), which is based on PRAAT, to estimate the formant frequencies. Regarding speech naturalness, we use the objective measure provided by NISQA (Mittag & Möller, 2020). This metric (the higher the better) was developed in the context of speech transformation algorithms and it was shown to highly correlate with subjective mean opinion scores (MOS) (i.e., human ratings). As a reference, the score provided by NISQA on the original dataset of English vowels (i.e., without any processing) is equal to $2.60 \pm 0.53$.

**Methods**    We compare the proposed approach with several methods from the literature: (i) Time-domain pitch-synchronous overlap-and-add (TD-PSOLA) (Moulines & Charpentier, 1990) performs fundamental frequency modifications through the decomposition of the signal into pitch-synchronized overlapping frames. (ii) WORLD (Morise et al., 2016) is a vocoder also used for fundamental-frequency modifications. It decomposes the speech signal into three components characterizing the fundamental frequency, the aperiodicity, and the spectral envelope. (iii) The method proposed by Hsu et al. (2017b) (here referred to as "VAE baseline") consists in applying translations directly in the latent space of the VAE. Unlike the proposed approach, this method requires predefined latent attribute representations $\boldsymbol{\mu}_{\text{src}}$ and $\boldsymbol{\mu}_{\text{trgt}}$ associated with the source and target values of the factor to be modified, respectively. In particular, computing $\boldsymbol{\mu}_{\text{src}}$ requires analyzing the input speech signal, for instance to estimate the fundamental frequency, which is not the case for the proposed method. The source and target latent attribute representations are then used to perform the translation $\tilde{\mathbf{z}} = \mathbf{z} - \boldsymbol{\mu}_{\text{src}} + \boldsymbol{\mu}_{\text{trgt}}$, where $\mathbf{z}$ and $\tilde{\mathbf{z}}$ are respectively the original and modified latent vectors. To ensure fair comparison, we build dictionaries of predefined latent attribute representations using the same artificially-generated speech data that were used in the weakly-supervised training stage of the proposed method. All the methods we compare with require a pre-processing of the input speech signal to compute the input trajectory of the factor to be modified (e.g. the fundamental frequency), which is not the case of the proposed method.

**Discussion**    Experimental results (mean and standard deviation) are shown in Table 1. Compared to the VAE baseline, the proposed method obtains better performance in terms of accuracy, disentanglement, and naturalness. These results confirm the effectiveness of performing the transformations in the learned latent subspaces and not directly in the latent space, as well as the advantage of using regression models instead of predefined latent attribute representations. Regarding $f_0$ transformation, WORLD obtains the best performance in terms of disentanglement, which is because the source and filter contributions are decoupled in the architecture of the vocoder. In terms of naturalness, WORLD and then TD-PSOLA obtain the best performance. This may be explained by the fact that these method operate directly in the time domain, therefore they do not suffer from phase reconstruction artifacts, unlike the proposed and VAE baseline methods. Naturalness is indeed greatly affected by phase reconstruction artificats, even from an unaltered speech spectrogram (i.e., without transformation). Phase reconstruction in a multi-speaker setting is still an open problem in speech processing. We want to emphasize that the objective of this study is not to compete with traditional signal processing methods such as TD-PSOLA and WORLD. It is rather to advance on the understanding of deep generative modeling of speech signals and to compare honestly with highly-specialized traditional systems. TD-PSOLA and WORLD exploit signal models specifically designed for the task at hand, which for instance prevents them to be used for modifying formant frequencies. In contrast, the proposed method is fully based on learning and the same methodology applies for modifying the fundamental or formant frequencies.

## 5  RELATED WORK

Variants of the VAE have recently led to considerable progress in disentangled representation learning (Kim & Mnih, 2018; Higgins et al., 2017; Chen et al., 2018). From experimental analyses on image data, these methods suggest that a vanilla VAE cannot learn a disentangled representation. In the present study, we experimentally show that learning a disentangled source-filter representation of speech using a simple VAE is possible, complying with the definition of disentanglement proposed in (Higgins et al., 2018). Locatello et al. (2019; 2020) recently showed both theoretically and from a large-scale experimental study that the unsupervised learning of disentangled representations

is impossible without inductive biases on both the models and the data. In the present study, we precisely employ a few examples of artificially-generated labeled speech data in order to disentangle the latent representation of a simple VAE, in terms of source-filter factors of speech variation.

Modifying speech signals characteristics such as the pitch or timbre is a highly covered research topic where deep generative modeling has recently emerged as a promising approach. Voice conversion consists in modifying speaker characteristics while keeping the linguistic information unchanged, e.g. (Kaneko & Kameoka, 2018; Kaneko et al., 2019; Kameoka et al., 2020). Many voice conversion methods use text information as input (Wang et al., 2018; Zhang et al., 2019; Kulkarni et al., 2021; Li et al., 2021), which greatly helps for transferring non-verbal vocal attributes from one speech signal to another, but it requires datasets labeled with text. Qian et al. (2020) and Webber et al. (2020) recently proposed speech transformation methods based on latent representation learning without resorting to textual information. The interpretability of the learned representations is enforced by the design of the model. Qian et al. (2020) proposed to use three independent encoder networks to decompose a speech signal into fundamental frequency, timbre and rhythm latent representations. Webber et al. (2020) focused on controlling source-filter parameters in speech signals, where the ability to control a given parameter (e.g. fundamental frequency) is enforced explicitly using labeled data and adversarial learning. In this approach, each parameter to be controlled requires a dedicated training of the model. Moreover, these methods are speaker-dependent, as speech generation in Qian et al. (2020) is conditioned on the speaker identity and Webber et al. (2020) used a single-speaker training dataset. This contrasts with the proposed method which is speaker-independent, and in which the source-filter representation of speech naturally emerges in the latent space of a single unsupervised VAE.

Hsu et al. (2017a;b) proposed to use VAEs for modifying the speaker identity and the phonemic content of speech signals. In (Hsu et al., 2017b), this is achieved by translations in the latent space. This is the baseline VAE approach we presented and compared with in Section 4. It requires to know predefined values of the latent representations associated with the source/target speech attributes to be modified. Conversely, Equation (8) in the proposed method does not require the knowledge of the factor $f_i$ associated with source vector $\mathbf{z}$, it only requires the value associated with the target vector $\tilde{\mathbf{z}}$. Moreover, Hsu et al. (2017b) did not address the control of continuous factors of speech variation in the VAE latent space, contrary to the present work.

Several methods have been recently proposed to control continuous factors of variation in deep generative models (Jahanian et al., 2019; Plumerault et al., 2020; Goetschalckx et al., 2019; Härkönen et al., 2020), focusing essentially on generative adversarial networks. They consist in identifying and then moving onto semantically meaningful directions in the latent space of the model. The present work is inspired by (Plumerault et al., 2020), which assumes that a factor of variation can be predicted from the projection of the latent vector along a specific axis, learned from artificially generated trajectories. The proposed method is however more generic, thanks to the learning of latent subspaces associated to the latent factors and to the introduction of a general formalism based on the use of "biased" aggregated posteriors. Moreover, these previous works on controlling deep generative models only allow for moving "blindly" onto semantically meaningful directions in the latent space. In the present study, we are able to generate data conditioned on a specific target value for a given factor of variation. Finally, previous work focused on image data, and to the best of our knowledge, this is the first approach to control source-filter factors of speech variation in a VAE.

## 6 CONCLUSION

In this work, using only a few seconds of artificially generated labeled speech data, we showed that the fundamental frequency and formant frequencies are encoded in orthogonal latent subspaces of an unsupervised VAE and we proposed a weakly-supervised method to control these attributes within the learned subspaces. The method generalizes well when applied to natural speech signals. To the best of our knowledge, this is the first approach that, with a single methodology, is able to extract, identify and control the source and filter low-level speech attributes within a VAE latent space. The proposed method could also be applied to other types of data such as natural or face images, provided that one can create a few synthetic images that capture variations in a single latent factor of interest, independently of others. For example, this would be possible for factors encoding the position, rotation and scale of objects in natural images, or some facial expressions.

## 7 REPRODUCIBILITY STATEMENT

To ensure reproducibility, the implementation of the method and the data generated with soundgen that are used to learn the latent subspaces and regression models will be made freely available online.

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

## A  SOLUTION TO THE LATENT SUBSPACE LEARNING PROBLEM

In this Appendix, we show that the solution to the optimization problem (5) is given by the principal eigenvectors of $\mathbf{S}_\phi(\mathcal{D}_i)$ in Equation (6).

Without loss of generality, we formulate the problem for a centered version of the latent data

$$\mathbf{z} \leftarrow \mathbf{z} - \boldsymbol{\mu}_\phi(\mathcal{D}_i), \tag{11}$$

where $\boldsymbol{\mu}_\phi(\mathcal{D}_i)$ is defined in Equation (4). This centering also affects the inference model originally defined in Equation (2), as follows:

$$q_\phi(\mathbf{z}|\mathbf{x}) = \mathcal{N}\left(\mathbf{z}; \boldsymbol{\mu}_\phi(\mathbf{x}) - \boldsymbol{\mu}_\phi(\mathcal{D}_i), \mathrm{diag}\{\mathbf{v}_\phi(\mathbf{x})\}\right). \tag{12}$$

Using Equation (3), the fact that $\mathbf{U}^\top \mathbf{U} = \mathbf{I}$, and Equation (12), the cost function in the optimization problem (5) can be rewritten as follows:

$$\mathbb{E}_{\hat{q}_\phi^{(i)}(\mathbf{z})}\left[\left\|\mathbf{z} - \mathbf{U}\mathbf{U}^\top\mathbf{z}\right\|_2^2\right] = \frac{1}{\#\mathcal{D}_i}\sum_{\mathbf{x}_n \in \mathcal{D}_i}\mathbb{E}_{q_\phi(\mathbf{z}|\mathbf{x}_n)}\left[\left\|\mathbf{z} - \mathbf{U}\mathbf{U}^\top\mathbf{z}\right\|_2^2\right]$$

$$= \mathrm{tr}\left\{(\mathbf{I} - \mathbf{U}\mathbf{U}^\top)\frac{1}{\#\mathcal{D}_i}\sum_{\mathbf{x}_n \in \mathcal{D}_i}\mathbb{E}_{q_\phi(\mathbf{z}|\mathbf{x}_n)}[\mathbf{z}\mathbf{z}^\top]\right\}$$

$$= \mathrm{tr}\left\{(\mathbf{I} - \mathbf{U}\mathbf{U}^\top)\mathbf{S}_\phi(\mathcal{D}_i)\right\}, \tag{13}$$

where $\mathbf{S}_\phi(\mathcal{D}_i)$ is defined in Equation (6). From this last equality, we see that the optimization problem (5) is equivalent to

$$\max_{\mathbf{U} \in \mathbb{R}^{L \times M_i}} \mathrm{tr}\left\{\mathbf{U}^\top \mathbf{S}_\phi(\mathcal{D}_i)\mathbf{U}\right\}, \qquad s.t. \ \mathbf{U}^\top\mathbf{U} = \mathbf{I}. \tag{14}$$

Very similarly to PCA (Pearson, 1901), the solution is given by the $M_i$ dominant eigenvectors of $\mathbf{S}_\phi(\mathcal{D}_i)$ (i.e., associated to the $M_i$ largest eigenvalues) (Bishop, 2006, Section 12.1).

## B  EXPERIMENTAL SETUP

### B.1  VAE TRAINING

To train the IS-VAE model (Bando et al., 2018; Leglaive et al., 2018; Girin et al., 2019), we use the Wall Street Journal (WSJ0) dataset (Garofolo et al., 1993a) which contains 25 hours of speech signals sampled at 16 kHz, including 52 female and 49 male speakers. The time-domain spech signals are converted to power spectrograms using the short-time Fourier transform (STFT) with an analysis window of length 64 ms and an overlap of 75%. The architecture of the VAE is shown in Figure 4, the encoder and decoder networks each have three dense layers. A hyperbolic tangent ($\tanh$) activation function is used at each layer, except for the output layers of the encoder and decoder where we use the identity function. The VAE input/output dimension is $D = 513$ (we only keep the non redundant part of the power spectrum at a given time frame) and the latent vector dimension is set to $L = 16$. We train the model using the Adam optimizer (Kingma & Ba, 2015) with a learning rate equal to 0.001.

### B.2  ANALYZING AND CONTROLLING SOURCE-FILTER FACTORS OF SPEECH VARIATION IN THE VAE

For a given factor of variation, the corresponding latent subspace is learned (see Section 3.1) using trajectories of speech power spectra generated with Soundgen (Anikin, 2019), all other factors being arbitrarily fixed. For $f_0$, the trajectory contains 226 points (which corresponds to 3.6 seconds of speech) evenly spaced in the range $[85, 310]$ Hz, $f_1$, $f_2$ and $f_3$ are fixed to 600 Hz, 1500 Hz, and 3200 Hz, respectively. For $f_1$, the trajectory contains 401 points (which corresponds to 6.4 seconds of speech) evenly spaced in the range $[200, 1000]$ Hz, $f_0$, $f_2$ and $f_3$ are fixed to 140 Hz, 1600 Hz, and 3200 Hz, respectively. For $f_2$, the trajectory contains 401 points evenly spaced in the range $[800, 2800]$ Hz, $f_0$, $f_1$ and $f_3$ are fixed to 140 Hz, 500 Hz, and 3200 Hz, respectively. For $f_3$, the trajectory contains 241 points (which corresponds to 3.9 seconds of speech) evenly spaced in the

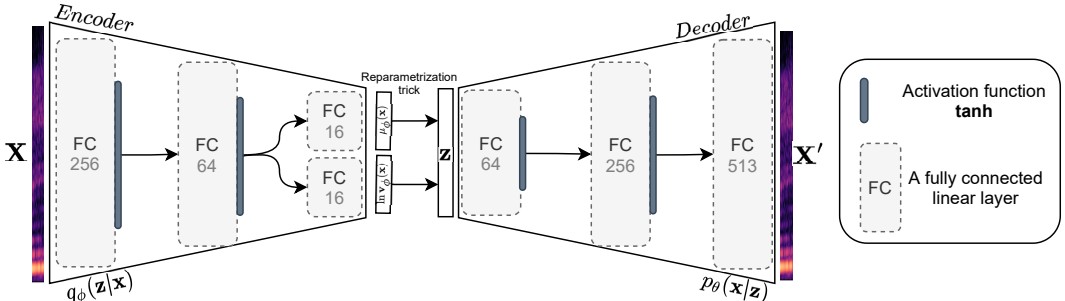

Figure 4: Architecture of the VAE. The number of neurons for each fully-connected layer is indicated below "FC", within each gray box. The input/output vector dimension of the VAE is 513 and the latent vector dimension is 16.

range $[2000, 3200]$ Hz, $f_0$, $f_1$ and $f_2$ are fixed to 140 Hz, 500 Hz, and 1200 Hz, respectively. The amplitude of the formants is set to 30 dB, and the bandwidth is automatically calculated from the fromant frequencies by Soundgen, using a formula derived from human phonetic research (Anikin, 2019). The regression models used to control the speech factors of variation in the latent space (see Sections 3.3) are learned on the same trajectories, but using the labels that correspond to the input control parameters of Soundgen.

## C    ADDITIONAL QUALITATIVE RESULTS

### C.1    VISUALIZATION OF THE LEARNED LATENT SUBSPACES

For $i = 0, 1, 2$ and $3$, Figures 5a, 5b, 5c, 5d are respectively obtained by projecting the latent mean vectors $\boldsymbol{\mu}_\phi(\mathbf{x}) \in \mathbb{R}^L$, for all data vectors $\mathbf{x} \in \mathcal{D}_i$, within the latent subspace characterized by $\mathbf{U}_i \in \mathbb{R}^{L \times M_i}$ (i.e., we perform dimensionality reduction). In the previously reported experiments, the latent subspace dimension $M_i$ of each factor of variation was chosen such that $80\%$ of the data variance was retained in the latent space. It resulted in $M_0 = 4$, $M_1 = 1$ and $M_2 = M_3 = 3$. In this section, for visualization purposes, we set $M_i = 3$ for all $i \in \{0, 1, 2, 3\}$. However, we can see that the $f_1$ trajectory (Figure 5b) is mainly concentrated along a single axis. Regarding $f_0$ (Figure 5a), setting $M_0 = 3$ retained $78\%$ of the variance of $\mathcal{D}_0$ in the latent space. An important observation that we make from these figures is that two data vectors $\mathbf{x}$ and $\mathbf{x}'$, corresponding to values of a given factor that are close, have projections of $\boldsymbol{\mu}_\phi(\mathbf{x})$ and $\boldsymbol{\mu}_\phi(\mathbf{x}')$ that are also close in the learned latent subspaces. This can be seen from the color bars which indicate the values of the factors of variation. It indicates that the learned representation preserves the notion of proximity in terms of fundamental frequency and formant frequencies.

In Figure 5e, we project three different datasets $\mathcal{D}_1$, defined for three different values of $f_2$. Similarly, in Figure 5f we show the trajectories associated with the projection of three datasets $\mathcal{D}_2$, defined for three different values of $f_1$. We notice that as expected, the trajectories are almost identical and only differ by a translation.

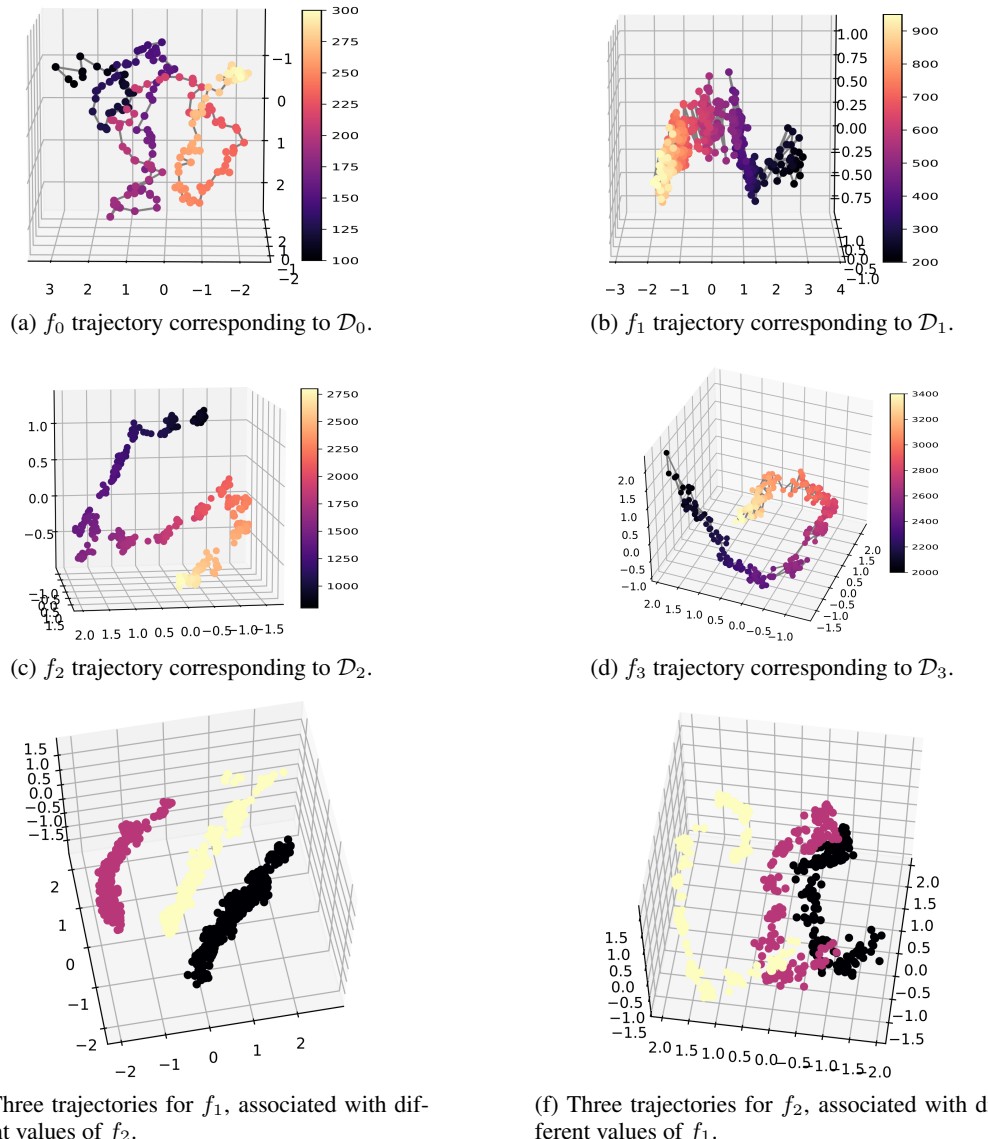

(a) $f_0$ trajectory corresponding to $\mathcal{D}_0$.

(b) $f_1$ trajectory corresponding to $\mathcal{D}_1$.

(c) $f_2$ trajectory corresponding to $\mathcal{D}_2$.

(d) $f_3$ trajectory corresponding to $\mathcal{D}_3$.

(e) Three trajectories for $f_1$, associated with different values of $f_2$.

(f) Three trajectories for $f_2$, associated with different values of $f_1$.

Figure 5: Visualization of trajectories in the learned latent subspaces.

## C.2 EXAMPLES OF GENERATED SPEECH SPECTRA

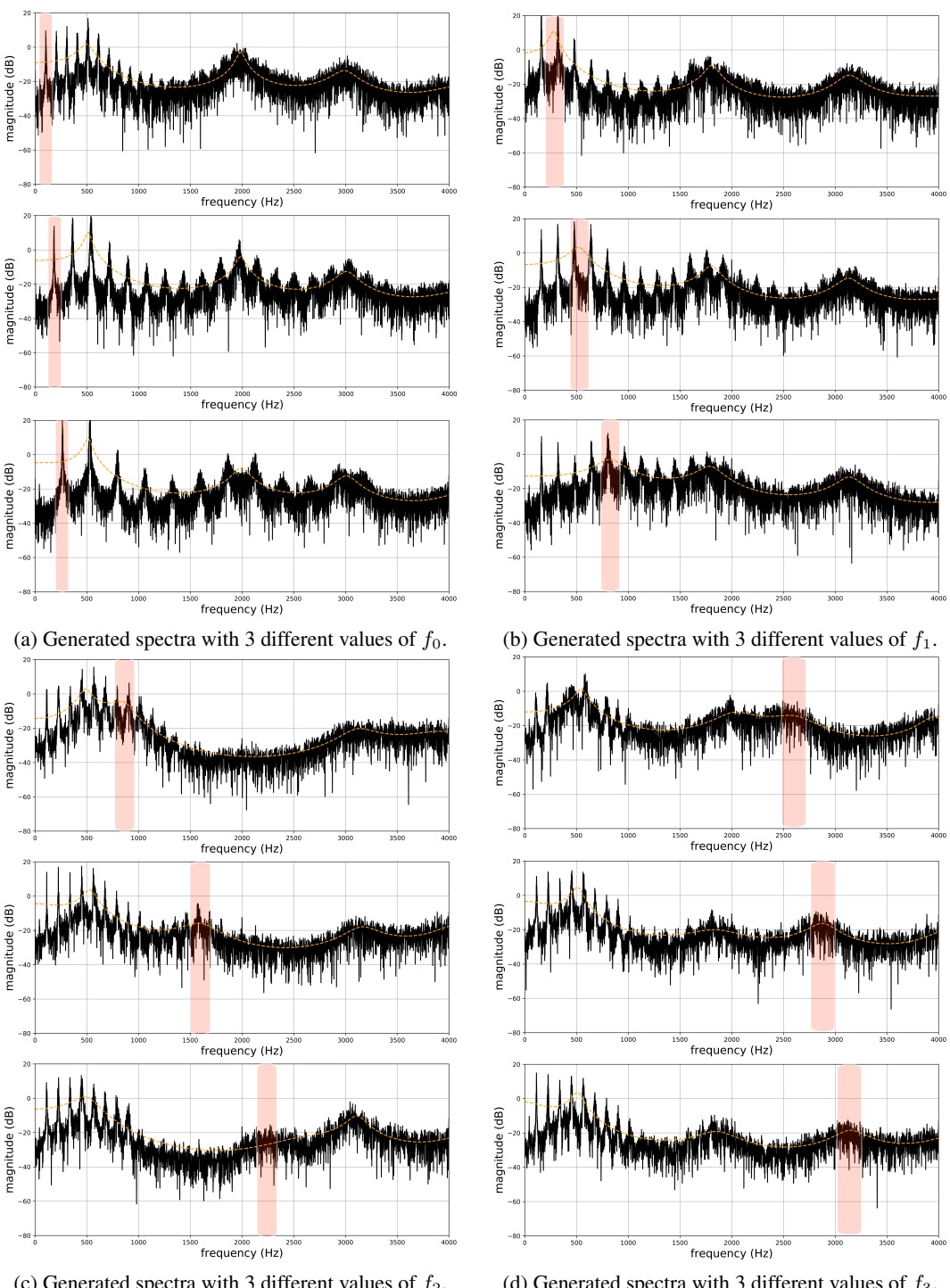

(a) Generated spectra with 3 different values of $f_0$.  (b) Generated spectra with 3 different values of $f_1$.

(c) Generated spectra with 3 different values of $f_2$.  (d) Generated spectra with 3 different values of $f_3$.

Figure 6: Power spectra (solid black line) and spectral envelopes (dashed orange line) obtained using the conditional prior in equation (9) (generalized to conditioning on multiple factors). Each subfigure contains three plots where we vary the value of one single factor at a time: $f_0$ in (a), $f_1$ in (b), $f_2$ in (c) and $f_3$ in (d).

## C.3 EXAMPLES OF TRANSFORMED SPEECH SPECTROGRAMS

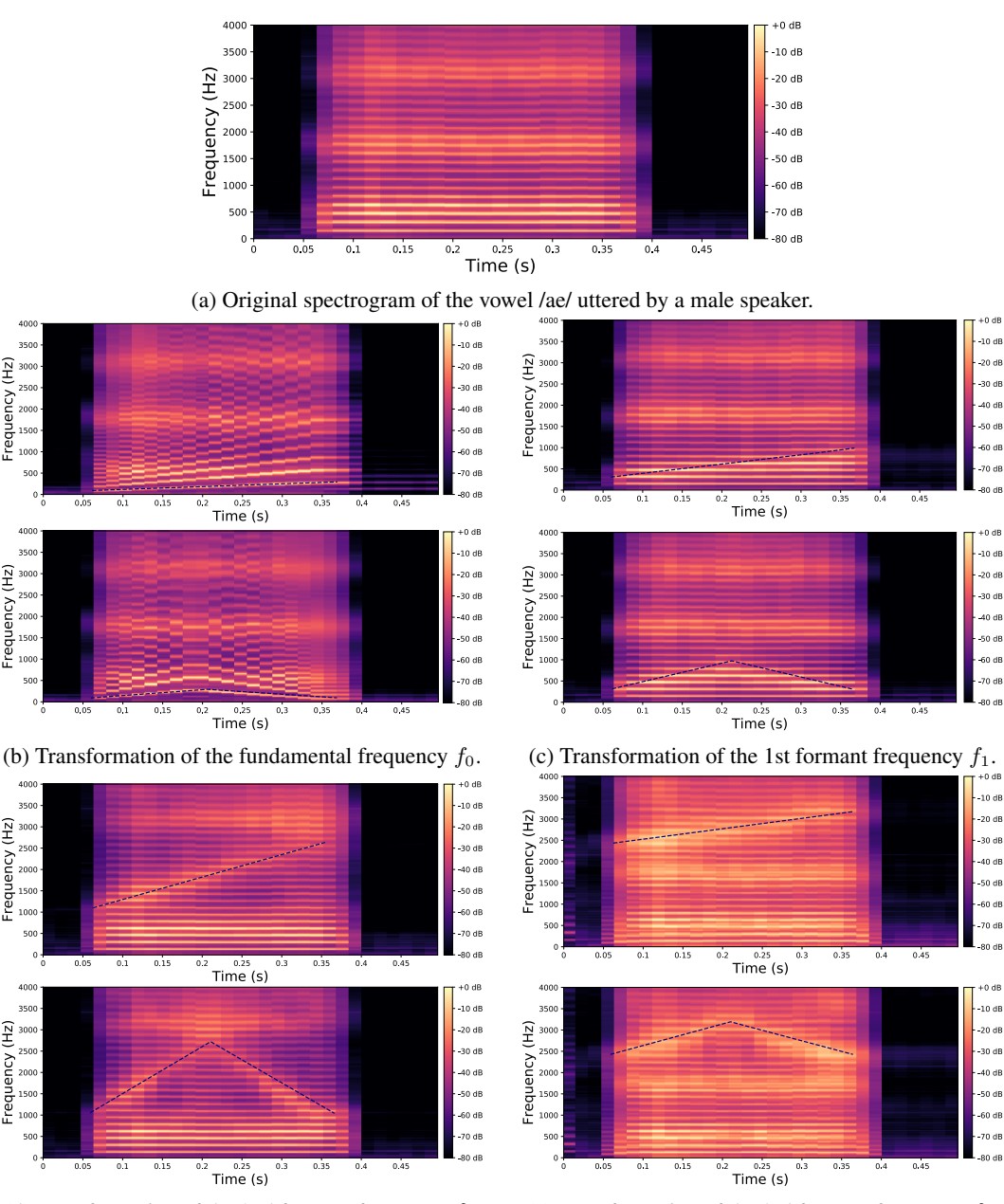

(a) Original spectrogram of the vowel /ae/ uttered by a male speaker.

(b) Transformation of the fundamental frequency $f_0$.     (c) Transformation of the 1st formant frequency $f_1$.

(d) Transformation of the 2nd formant frequency $f_2$.     (e) Transformation of the 3rd formant frequency $f_3$.

Figure 7: Figure (a) shows the spectrogram of a vowel uttered by a male speaker. Figures (b), (c), (d) and (e) show transformations of this spectrogram with the proposed method, where we vary $f_0$, $f_1$, $f_2$, and $f_3$, respectively. The target value for these factors is indicated by the dashed blue line.

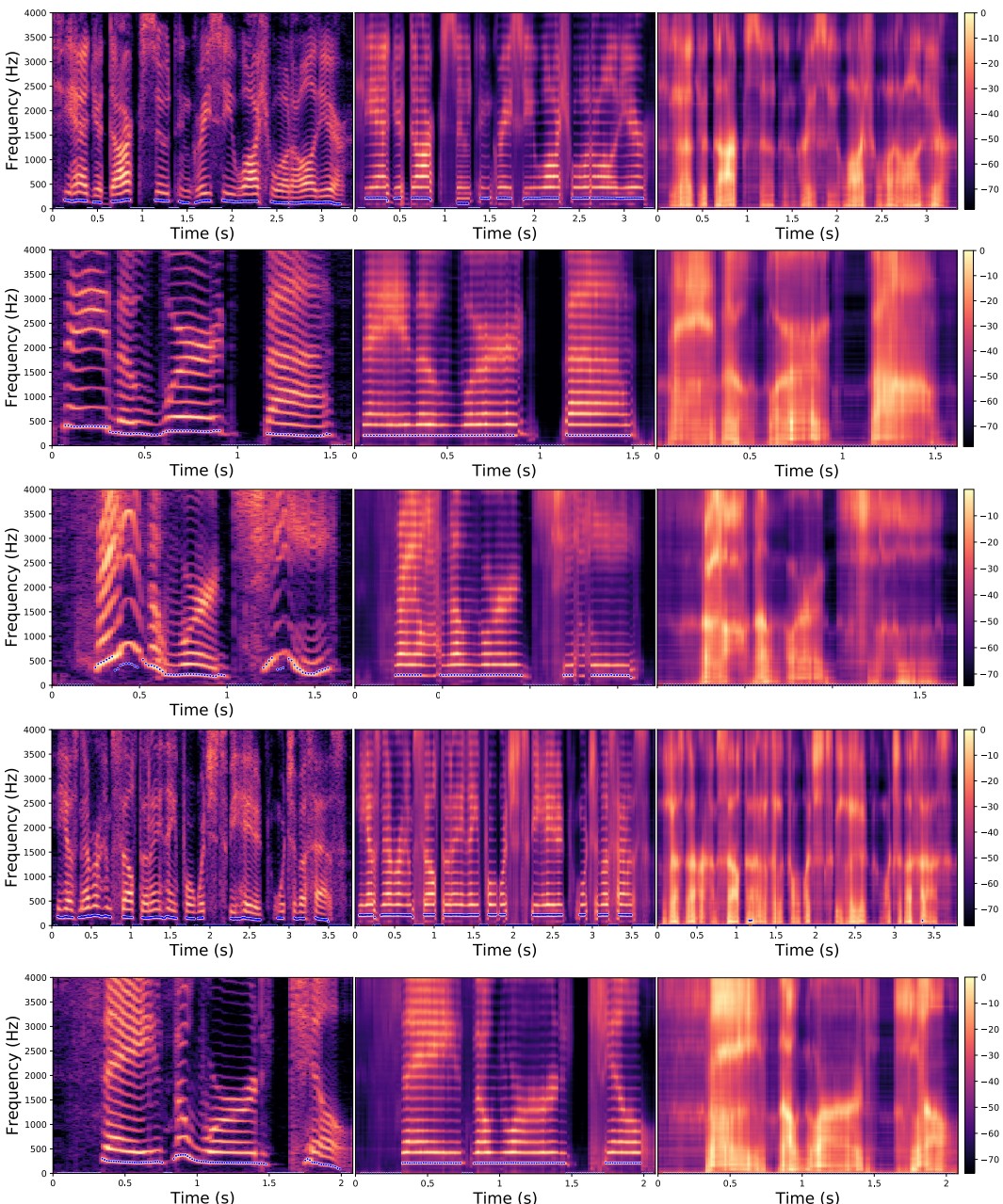

Figure 8: Each line in this figure corresponds to a speech signal uttered by a different speaker. Left: spectrogram of the original speech signal; Middle: transformed spectrogram where the fundamental frequency is set constant over time; Right: transformed spectrogram where the original voiced speech signal (left) is converted into a whispered speech signal (i.e., the fundamental frequency is removed).

# D ADDITIONAL QUANTITATIVE RESULTS

## D.1 TRANSFORMATIONS OF THE THIRD FORMANT FREQUENCY ON THE ENGLISH VOWELS DATASET

Table 2 completes Table 1 by including transformations of the third formant frequency in the range $[2200, 3200]$ Hz, with a step of 20 Hz. The new column $\delta f_3$ also measures the performance in terms of disentanglement when modifying other factors $f_i \neq f_3$. These additional results do not modify the conclusions drawn in Section 4.

Table 2: Performance (mean and std) for the fundamental frequency ($f_0$) and formant frequencies ($f_1$, $f_2$ and $f_3$) transformations of English vowels.

| Factor | Method | NISQA $(\uparrow)$ | $\delta f_0$ $(\%, \downarrow)$ | $\delta f_1$ $(\%, \downarrow)$ | $\delta f_2$ $(\%, \downarrow)$ | $\delta f_3$ $(\%, \downarrow)$ |
|---|---|---|---|---|---|---|
| $f_0$ | TD-PSOLA | 2.32 $\pm$ 0.55 | 3.8 $\pm$ 2.5 | 6.3 $\pm$ 2.8 | 3.7 $\pm$ 0.9 | 2.1 $\pm$ 0.5 |
| | WORLD | 2.49 $\pm$ 0.60 | 4.5 $\pm$ 0.6 | 3.7 $\pm$ 1.8 | 2.3 $\pm$ 0.7 | 1.2 $\pm$ 0.2 |
| | VAE baseline | 1.94 $\pm$ 0.43 | 6.21 $\pm$ 2.8 | 10.4 $\pm$ 2.4 | 6.2 $\pm$ 0.9 | 4.5 $\pm$ 0.2 |
| | Proposed | 2.08 $\pm$ 0.48 | 0.8 $\pm$ 0.2 | 7.2 $\pm$ 1.3 | 3.6 $\pm$ 1.2 | 3.8 $\pm$ 0.3 |
| $f_1$ | VAE baseline | 1.84 $\pm$ 0.5 | 11.3 $\pm$ 4.2 | 15.1 $\pm$ 3.5 | 6.0 $\pm$ 1.2 | 4.2 $\pm$ 0.4 |
| | Proposed | 1.85 $\pm$ 0.4 | 6.0 $\pm$ 1.6 | 8.4 $\pm$ 3.2 | 5.7 $\pm$ 0.4 | 4.4 $\pm$ 0.3 |
| $f_2$ | VAE baseline | 2.01 $\pm$ 0.4 | 19.5 $\pm$ 3.2 | 10.7 $\pm$ 0.5 | 10.9 $\pm$ 1.9 | 5.8 $\pm$ 0.6 |
| | Proposed | 2.03 $\pm$ 0.43 | 8.5 $\pm$ 1.1 | 8.7 $\pm$ 1.1 | 6.2 $\pm$ 1.5 | 5.8 $\pm$ 0.2 |
| $f_3$ | VAE baseline | 1.82 $\pm$ 0.14 | 27.0 $\pm$ 1.5 | 13.0 $\pm$ 1.3 | 12.0 $\pm$ 1.8 | 7.3 $\pm$ 1.5 |
| | Proposed | 1.94 $\pm$ 0.48 | 8.3 $\pm$ 1.0 | 8.6 $\pm$ 0.7 | 4.9 $\pm$ 0.9 | 2.0 $\pm$ 0.4 |

## D.2 ROBUSTNESS WITH RESPECT TO DIFFERENT TRAINING DATASETS FOR THE VAE

This Section investigates the robustness of the proposed method with respect to different datasets used to train the VAE model. This table presents results for modifications of the fundamental frequency only, applied to the English vowels dataset. We considered the following training datasets:

- WSJ: the Wall street Journal (Garofolo et al., 1993a) dataset that was used in the previous experiments.

- SIWIS: the SIWIS French speech synthesis dataset (Honnet et al., 2017), which contains more than ten hours of French speech recordings.

- TESS: the Toronto emotional speech dataset (Dupuis & Pichora-Fuller, 2010), which contains 2800 utterances spoken by two actresses using different emotions (anger, disgust, fear, happiness, pleasant surprise, sadness, and neutral).

- LJspeech: the LJspeech dataset (Ito & Johnson, 2017), which contains 13100 short audio clips of a single speaker reading passages from 7 non-fiction books.

The artificially-generated speech dataset used for learning the latent subspaces and the regression models along with the test dataset of English vowels remain the same. It can be seen in Table 3 that the performance remains quite stable with different VAE training datasets. WSJ is the largest dataset and therefore obtains the best performance. Interestingly, the results obtained with the SIWIS dataset of French speech signals remain satisfactory, even if there is a mismatch between the training (French) and testing (English) datasets.

Table 3: Performance (mean and std) for fundamental frequency ($f_0$) transformations of English vowels using different datasets for training the unsupervised VAE model.

| Dataset | NISQA $_{(\uparrow)}$ | $\delta f_0$ $_{(\%, \downarrow)}$ | $\delta f_1$ $_{(\%, \downarrow)}$ | $\delta f_2$ $_{(\%, \downarrow)}$ | $\delta f_3$ $_{(\%, \downarrow)}$ |
|---|---|---|---|---|---|
| WSJ | 2.08 $\pm 0.48$ | 0.8 $\pm 0.2$ | 7.2 $\pm 1.3$ | 3.6 $\pm 1.2$ | 3.8 $\pm 0.3$ |
| SIWIS | 1.93 $\pm 0.43$ | 1.2 $\pm 0.5$ | 10.0 $\pm 4.2$ | 8.3 $\pm 1.1$ | 14.0 $\pm 0.2$ |
| TESS | 1.98 $\pm 0.50$ | 2.7 $\pm 2.3$ | 9.3 $\pm 3.5$ | 9.0 $\pm 0.8$ | 7.0 $\pm 0.2$ |
| LJspeech | 1.96 $\pm 0.40$ | 1.2 $\pm 0.6$ | 9.3 $\pm 1.2$ | 5.6 $\pm 0.6$ | 4.6 $\pm 0.1$ |

### D.3 EVALUATION ON THE TIMIT DATASET

In this Section, we evaluate the proposed method on the TIMIT dataset (Garofolo et al., 1993b), using the VAE trained on the WSJ dataset. TIMIT is a corpus of phonemically and lexically transcribed speech of American English speakers of different sexes and dialects. We used the test corpus containing 1680 utterances. Because we are interested in studying the interaction between modifications of the fundamental frequency and formant frequencies, we only evaluate the method on the phonemes that are voiced (40 phonemes over a total of 52), which can be identified using the annotations. The ranges of transformation for the fundamental frequency, first, second and third formant frequencies are respectively $[100, 300]$ Hz, $[300, 900]$ Hz, $[1100, 2700]$ Hz, and $[2200, 3200]$ Hz with a step of 10 Hz, 50 Hz, 100 Hz and 50 Hz, respectively. The metrics and evaluated methods remain the same as those described in Section 4.2.2.

TIMIT is phonemically richer than the English vowels dataset previously used, however it is not labeled with the fundamental and formant frequencies. Therefore, we do not have the ground truth values which makes the evaluation more difficult than with the English vowels dataset. Instead of the ground truth, we use the formant frequencies and the fundamental frequency computed on the original speech utterances (i.e., before transformation) using the same tools as described in Section 4.2.2. This makes the evaluation on TIMIT less reliable than on the English vowels dataset, but it allows us to test the methods on a larger variety of phonemes.

Results are presented in Table 4. They are very consistent with the ones obtained on the English vowels dataset (see Tables 1 and 2), which further strengthens the conclusions drawn in the discussion paragraph of Section 4.2.2.

Table 4: Performance (mean and std) for the fundamental frequency ($f_0$) and formant frequencies ($f_1$, $f_2$ and $f_3$) transformations on the TIMIT dataset.

| Factor | Method | NISQA $_{(\uparrow)}$ | $\delta f_0$ $_{(\%, \downarrow)}$ | $\delta f_1$ $_{(\%, \downarrow)}$ | $\delta f_2$ $_{(\%, \downarrow)}$ | $\delta f_3$ $_{(\%, \downarrow)}$ |
|---|---|---|---|---|---|---|
| $f_0$ | TD-PSOLA | 2.36 $\pm 0.50$ | 2.4 $\pm 1.9$ | 7.9 $\pm 0.6$ | 4.5 $\pm 0.3$ | 3.9 $\pm 0.2$ |
|  | WORLD | 2.45 $\pm 0.47$ | 0.3 $\pm 0.1$ | 7.1 $\pm 1.2$ | 6.2 $\pm 0.4$ | 4.2 $\pm 0.2$ |
|  | VAE baseline | 1.59 $\pm 0.43$ | 16.1 $\pm 6.3$ | 17.0 $\pm 3.0$ | 12.1 $\pm 0.2$ | 10.9 $\pm 1.3$ |
|  | Proposed | 2.28 $\pm 0.57$ | 0.8 $\pm 0.6$ | 9.1 $\pm 1.1$ | 8.3 $\pm 0.9$ | 6.0 $\pm 1.8$ |
| $f_1$ | VAE baseline | 1.42 $\pm 0.34$ | 10.1 $\pm 2.8$ | 16.4 $\pm 1.4$ | 12.4 $\pm 0.9$ | 11.2 $\pm 2.6$ |
|  | Proposed | 1.66 $\pm 0.31$ | 7.1 $\pm 3.6$ | 9.2 $\pm 0.8$ | 9.0 $\pm 1.3$ | 7.8 $\pm 1.1$ |
| $f_2$ | VAE baseline | 1.46 $\pm 0.30$ | 19.3 $\pm 5.0$ | 16.4 $\pm 0.8$ | 20.3 $\pm 6.3$ | 11.5 $\pm 0.5$ |
|  | Proposed | 1.49 $\pm 0.30$ | 9.1 $\pm 2.2$ | 8.3 $\pm 1.3$ | 4.3 $\pm 1.3$ | 8.1 $\pm 0.2$ |
| $f_3$ | VAE baseline | 1.40 $\pm 0.48$ | 20.4 $\pm 1.0$ | 17.4 $\pm 0.2$ | 14.4 $\pm 0.2$ | 11.7 $\pm 2.3$ |
|  | Proposed | 1.48 $\pm 0.42$ | 8.5 $\pm 1.9$ | 8.7 $\pm 0.9$ | 5.7 $\pm 2.1$ | 2.5 $\pm 1.8$ |

