# OpenReview forum: "Learning and controlling the source-filter representation of speech with a variational autoencoder"
_ICLR.cc/2022/Conference — ICLR 2022 Submitted_

### Official Review · Reviewer_vTAw · 2021-10-28

**Correctness:** 4
**Technical Novelty And Significance:** 3
**Empirical Novelty And Significance:** 2
**Recommendation:** 6
**Confidence:** 4

**Main Review:**

strengths:

The fact that one can control the fundamental frequency and formant frequencies of speech by manipulating the latent space is an interesting finding.
Finding the subspaces of latent the VAE latent space using artificially controllable dataset is an interesting idea.


weaknesses:

It's interesting work to someone who is working in the speech signal processing field.
However, I wonder how much implication this work could give to general audiences.
The fact that the VAE latent space can be manipulated using an artificially controllable dataset is interesting but at the same time, it is a disadvantage because we cannot find the subspaces if the artificially controllable dataset cannot be acquired.


**Summary Of The Paper:**

This paper shows that the fundamental frequency and formant frequency information is encoded in a speech VAE model.
This can be found by using artificially controlled/generated dataset.
After finding how to manipulate the latent space, one can control arbitrary speech samples in a desirable way, such as controlling fundamental frequency or formant frequencies.
The authors also show that the harmonic parts can be removed from the original speech and reconstructed as a whispered voice using only the spectral envelope part.
Experiment results show that the proposed method can control formant frequencies and show that it can control better than the method proposed by previous work.
The proposed method show on par or worse performance on fundamental frequency control experiment when compared to traditional DSP vocoders, such as TD-PSOLA or WORLD.


**Summary Of The Review:**

This paper shows interesting findings on speech VAE.
The proposed method is straightforward and I think it can draw some attention from the speech signal processing field.
I recommend [6: marginally above the acceptance threshold].

Questions:
1. Why was IS-VAE chosen to model speech?
2. On Page 3, Section 3, second paragraph, "Di denote a dataset of artificially- generated speech vectors (more precisely short-term power spectra) synthesized by varying only fi, all other factors {fj , j ̸= i} being arbitrarily fixed", does this mean that when making D_0, f1, f2, and f3 is fixed? If that's true what values for f1, f2, and f3 are chosen?
3. Can we still find subspaces when we do not have the artificial generator? (e.g., by manually selecting speech samples we would like to take control of?)

References:
Some papers the authors can consider to add in the reference:
[1] shows that one can disentangle source and filter in a supervised manner.
Recently [2] showed that the source and filter parts can be disentangled even in a self-supervised setting.

[1] https://arxiv.org/abs/1908.01919
[2] https://arxiv.org/abs/2110.14513

---

> ### Author Response · Authors · 2021-11-18
> **Response to Reviewer vTAw (Part 2)**
>
> > **Why was IS-VAE chosen to model speech ?**
>
> We are interested in modeling speech spectrograms, so we have to design a generative model dedicated to non-negative data. This requirement prevents us from using the standard Gaussian VAE as in e.g. image modeling. The IS-VAE model is a popular choice because it also corresponds to a generative model of the complex-valued STFT coefficient of the speech signals. Each STFT coefficient is modeled as a circularly-symmetric complex Gaussian random variable with a variance that corresponds to the variable $v_\theta(\mathbf{z})$ in Equation (1), see e.g. Girin et al. (2019). This choice of the IS-VAE model is also supported by previous works on generative modeling of audio signals with nonnegative matrix factorization techniques, which have been very popular for the past decade, see for instance C. Févotte et al., “Nonnegative matrix factorization with the Itakura-Saito divergence: With application to music analysis”, Neural computation, 2009. Using the IS-VAE model also draws interesting perspectives regarding the use of the proposed method to develop pitch-informed extensions of speech enhancement and separation methods also using the IS-VAE model (Bando et al., 2018; Leglaive et al., 2018; Kameoka et al., 2019).
>
>
> > **On Page 3, Section 3, second paragraph, "Di denote a dataset of artificially- generated speech vectors (more precisely short-term power spectra) synthesized by varying only fi, all other factors {fj , j ̸= i} being arbitrarily fixed", does this mean that when making D_0, f1, f2, and f3 is fixed? If that's true what values for f1, f2, and f3 are chosen?**
>
> Indeed, when creating D_0,  f1, f2, and f3 were arbitrarily fixed. All values were provided in Appendix B.2 of the original submission. We quote the corresponding paragraph below to answer the question of the Reviewer:
>
> “For f0, the trajectory contains 226 points evenly spaced in the range [85, 310] Hz, f1, f2 and f3 are fixed to 600 Hz, 1500 Hz, and 3200 Hz, respectively. For f1, the trajectory contains 401 points evenly spaced in the range [200, 1000] Hz, f0, f2 and f3 are fixed to 140 Hz, 1600 Hz, and 3200 Hz, respectively. For f2, the trajectory contains 401 points evenly spaced in the range [800, 2800] Hz, f0, f1 and f3 are fixed to 140 Hz, 500 Hz, and 3200 Hz, respectively. For f3, the trajectory contains 241 points evenly spaced in the range [2000, 3200] Hz, f0, f1 and f2 are fixed to 140 Hz, 500 Hz, and 1200 Hz, respectively. ”
>
>
> > **3. Can we still find subspaces when we do not have the artificial generator? (e.g., by manually selecting speech samples we would like to take control of ?**
>
> This is a very interesting question. A key requirement of the method is to be able to generate a very few examples that capture variations in a single factor/attribute, independently of other ones that should remain fixed.  We reached this requirement using an artificial speech synthesizer, but alternatives indeed exist. The easiest one would be to record a few seconds of speech where a speaker varies one single factor at a time. This would be particularly easy for the fundamental frequency, as varying this factor while keeping the formant frequencies unaltered is quite easy for humans. As suggested by the Reviewer, another possibility would be to manually select speech samples in existing datasets, but it may be more difficult to find samples where only one factor varies.
>
>
> > **References: Some papers the authors can consider to add in the reference: [1] shows that one can disentangle source and filter in a supervised manner. Recently [2] showed that the source and filter parts can be disentangled even in a self-supervised setting.
> [1] https://arxiv.org/abs/1908.01919 [2] https://arxiv.org/abs/2110.14513**
>
> We thank the Reviewer for this suggestion. We added these references in the revised manuscript.

---

> > ### Comment · Reviewer_vTAw · 2021-11-23
> > **Keeping the score**
> >
> > The authors have answered all my questions.
> > I still doubt what aspects of speech can we control by the proposed method other than f0 though.
> > Usually the factor we would like to control is not always controllable, so I wonder what we can do with this approach when we have arbitrary attributes we would like to control. I imagined of manually selecting the samples, but I doubt this method will work for unquantifiable attributes.

---

> ### Author Response · Authors · 2021-11-18
> **Response to Reviewer vTAw (Part 1)**
>
> > **It's interesting work to someone who is working in the speech signal processing field. However, I wonder how much implication this work could give to general audiences.**
>
> Interpretation and control of learned representations with deep generative models (especially VAEs and GANs) is currently an important topic in the machine learning community, regardless of the application domain, and as confirmed by the list of relevant topics at ICLR. Previous works (see Related Work section of the submission) have mainly focused on natural images, and most of them were published in machine learning and not computer vision conferences. We would also like to emphasize that this submission matches with multiple items of the list of relevant topics indicated in the ICLR 2022 call for papers (https://iclr.cc/Conferences/2022/CallForPapers), namely “unsupervised, semi-supervised, and supervised representation learning”; “visualization or interpretation of learned representations”; “applications in audio, speech, robotics, neuroscience, computational biology, or any other field”. So as long as the interest of the audience matches with the above mentioned list of relevant topics, we believe that this submission is relevant for ICLR.
>
>
> > **The fact that the VAE latent space can be manipulated using an artificially controllable dataset is interesting but at the same time, it is a disadvantage because we cannot find the subspaces if the artificially controllable dataset cannot be acquired.**
>
> The proposed method might be applied to different domains such as image processing. For instance, it might be used for translating, rotating and scaling objects in natural images, or to adjust brightness. On face images, it could be used to perform expressive transformations, such as making faces smile. As pointed out by the reviewer, a key requirement would be to be able to generate image examples that capture variations in a single attribute, independently of others. We believe that creating synthetic images (e.g. images with a foreground object manually translated, rotated or scaled, or face images with someone progressively smiling) is not a difficult task. We also want to emphasize that the proposed method, when applied to speech data, required only a few synthetic examples to learn the speech attributes latent subspaces and corresponding regression models. In our experiments, we used trajectories of about 200 to 400 examples to learn the subspace and regression model of each speech attribute. This corresponds to between 3.2 and 6.4 seconds of controlled labeled speech data, to be compared with the 25 hours of unlabeled speech data used in the unsupervised training of the VAE (Wall Street Journal dataset). We could probably quite easily create a few hundred images so as to control the latent space of a VAE trained on a large-scale unlabeled natural or face image dataset.  Due to space constraints, we have only briefly discussed applications of the proposed method to image data in the conclusion of the revised manuscript.

---

### Official Review · Reviewer_e7Cb · 2021-11-01

**Correctness:** 4
**Technical Novelty And Significance:** 2
**Empirical Novelty And Significance:** 2
**Recommendation:** 5
**Confidence:** 4

**Main Review:**

Strengths
- This paper could be seen as an extension of Hsu et al. (2017a), both of which found that unsupervised VAEs learn to model speech properties such as formants in orthogonal subspaces. To transform speech, both consider an affine transformation that subtracts the component of the original property ($U_i U_i^T z$ in this paper and $\mu_{src}$ in Hsu et al. (2017)) and then add the component of the target property ($U_i g_{\eta_i} (y)$ in this paper and $\mu_{tgt}$ in Hsu et al. (2017)). There are two methodological improvements in this paper. First, the authors find the subspace modeling for each property U using labeled data, such that the subtraction can be done for an unlabeled instance. Second, a regression model is built to predict the coefficients of the target value for each property, such that this method can in principle generalize to unseen values for a property.
- Empirically, the proposed method demonstrates better source and filter parameter control compared to the VAE baseline (Hsu et al., 2017).

Weaknesses
- The main contribution is determining latent subspaces for F0 and formants and building a regression model to predict the coefficients of the components for each subspace. It offers limited real world applications since the performance still lags behind traditional signal-processing based methods if these attributes (source and filter parameters) were to be modified. The novelty is also limited and might have limited interest for the machine learning community. Specifically, the orthogonal subspace property has been shown in computer vision and speech modeling, and the authors have only demonstrated that the proposed method can be used to control a limited set of speech properties.


**Summary Of The Paper:**

This paper analyzes the latent representations of speech spectrograms learned by unsupervised variational autoencoders (VAEs) and discovers that the VAE learns to model the variation of fundamental frequencies (F0/source) and formant frequencies (filters) using orthogonal subspaces. Based on the discovery, the authors further propose to build a regression model that maps target value of a property to the factor loading coefficients of the subspace corresponding to that property, which enable control of properties independently.

**Summary Of The Review:**

This paper presents a nice extension from the literature. However, the novelty and applicability is limited and the generality of the proposed method for controlling other attributes is unclear. Hence, this could be of limited interest to the audience of the conference

---

> ### Author Response · Authors · 2021-11-18
> **Response to Reviewer e7Cb**
>
> > **This paper could be seen as an extension of Hsu et al. (2017a), both of which found that unsupervised VAEs learn to model speech properties such as formants in orthogonal subspaces.**
>
> We believe that the technical novelty of the proposed method as compared with Hsu et al. (2017a) is substantial. Hsu et al. (2017a) characterizes the latent space of a VAE in terms of discrete phoneme classes and speaker identities, while the proposed method focuses on continuous-valued factors corresponding to the fundamental and formant frequencies. Hsu et al. (2017a) only manipulates latent factors through interpolations, without control of the value of these factors. Controlling continuous latent factors in the generative process of speech data (as in the present paper) is much more challenging. Also, Hsu et al. (2017a) only shows that the latent representations of different phonemes are approximately orthogonal, and from this observation the authors arguably conclude that the “latent phone representations reside in orthogonal latent subspaces”. The paper by Hsu et al. (2017a) does not characterize these subspaces, contrary to the present paper that proposes a method to learn the basis matrices of the subspaces. This is a key feature to be able to move in the corresponding subspaces so as to perform disentangled and controlled speech transformations, as achieved in the proposed method using non-linear regression models from the factor to be controlled to the speech data coordinates in the corresponding learned subspace.
>
>
> > **It offers limited real world applications since the performance still lags behind traditional signal-processing based methods if these attributes (source and filter parameters) were to be modified.**
>
> As mentioned in the discussion of the experimental results, the objective of this paper is not to compete with traditional signal processing methods. It is rather to advance on the understanding of deep generative modeling of speech signals and to compare honestly with highly-specialized traditional signal processing systems. We believe that this objective is well aligned with the scope of ICLR as detailed in the call for papers. Moreover, there is currently a high interest in the use of VAEs for speech modeling and synthesis. Showing that a VAE naturally disentangles the source-filter structure offers several new possibilities, for instance to control the melodic aspect of the generated speech data, or to guide (with pitch information) speech enhancement, separation or more generally restoration methods that use VAE-based speech priors.
>
>
> > **The novelty is also limited and might have limited interest for the machine learning community.**
>
> Interpretation and control of learned representations with deep generative models (especially VAEs and GANs) is currently an important topic in the machine learning community, regardless of the application domain, and as confirmed by the list of relevant topics at ICLR. Previous works (see Related Work section of the submission) have mainly focused on natural images, and most of them were published in machine learning and not computer vision conferences.
>
>
> > **Specifically, the orthogonal subspace property has been shown in computer vision and speech modeling,**
>
> Regarding speech modeling, as detailed above, the fact that the fundamental and formant frequencies are encoded in orthogonal latent subspaces of an unsupervised VAE was not shown in Hsu et al. (2017a) or any other previous work, to the best of our knowledge. Regarding computer vision applications, could the Reviewer please be more specific about what he/she means by “the orthogonal subspace property has been shown”?
>
>
> > **and the authors have only demonstrated that the proposed method can be used to control a limited set of speech properties.**
>
> The set of speech properties is indeed limited, but these properties are fundamental in speech and by controlling them we can change various higher level properties related for instance to the expressivity or speaker identity.
>
>
> > **Hence, this could be of limited interest to the audience of the conference**
>
> We would like to emphasize that this submission matches with multiple items of the list of relevant topics indicated in the ICLR 2022 call for papers (https://iclr.cc/Conferences/2022/CallForPapers), namely “unsupervised, semi-supervised, and supervised representation learning”; “visualization or interpretation of learned representations”; “applications in audio, speech, robotics, neuroscience, computational biology, or any other field”. So as long as the interest of the audience matches with the above mentioned list of relevant topics, we believe that this submission is relevant for ICLR.

---

> > ### Comment · Reviewer_e7Cb · 2021-11-29
> > **Response to the rebuttal**
> >
> > Thank the authors for providing detailed responses to my comments. Here are a few follow-up comments
> >
> > > could the Reviewer please be more specific about what he/she means by “the orthogonal subspace property has been shown”?
> >
> > Dimension-wise disentanglement is a special case of the orthogonal subspace property, which have been demonstrated on (beta)-VAE [1, 2] on 3D Chairs / dSprites / CelebA. On the other hand, interpolating latent codes without changing the attributes common to the source on both ends, such as [3], also implies the orthogonality property implicitly as the translation between two codes controls rendering of only a specific attribute.
> >
> > [1] Higgins, Irina, et al. "beta-vae: Learning basic visual concepts with a constrained variational framework." (2016).
> > [2] Burgess, Christopher P., et al. "Understanding disentangling in $\beta $-VAE." arXiv preprint arXiv:1804.03599 (2018).
> > [3] Berthelot, David, et al. "Understanding and improving interpolation in autoencoders via an adversarial regularizer." arXiv preprint arXiv:1807.07543 (2018).
> >
> >
> > > We would like to emphasize that this submission matches with multiple items of the list of relevant topics
> >
> > Considering it as an application paper, it is still of limited interest. It is unclear or not demonstrated how this would useful for real world applications. If the purpose is to control source and filter parameters, it would be nice to demonstrate why such methods would be more favorable than signal processing based method, such as being more robust to noise and reverberation. The finding of being able to modify only a specific attribute in the latent space is not novel either, as it has been demonstrated in the reference mentioned above. The experiments were only conducted on a clean speech dataset, and whether such approach would be transferrable to other acoustic domains or even other domains (e.g., images) remains unverified.

---

### Official Review · Reviewer_kWcG · 2021-11-02

**Correctness:** 4
**Technical Novelty And Significance:** 2
**Empirical Novelty And Significance:** 2
**Recommendation:** 5
**Confidence:** 4

**Main Review:**

Strengths:
 - Clear and well written introduction to the source-filter model of speech production.
 - Clever method for identifying and controlling disentangled subspaces given supervised labels for them.
 - Convincing quantitative and qualitative demonstration in Section 4 that the proposed approach 1) learns a representation which is well disentangled and 2) can independently control each component.

Weaknesses:
 - Experiments are all on a relatively simple domain of speech spectrogram frames (in fact, restricted only to vowels, speech sounds which are most easily modeled using a source-filter decomposition).  The quantitative results (sec 4.2) are further restricted to *isolated* vowels, more closely matching the domain of the labeled synthetic data and not covering the dynamic space of variation in natural speech.
   - At the very least the paper could use a detailed discussion about the how the proposed approach might be applied to other domains of interest to the generative modeling community, e.g., to natural images or at least to more complex speech tasks.
     Such generalizations seems difficult since a key requirement for the technique to work is the existence of datasets which capture variation in a single attribute independently of others (or a synthesizer/generator which can create such datasets), i.e., it requires data which explicitly demonstrates disentanglement to use as supervision.  This requirement seems like it would be difficult to fulfill in other less restrictive domains.

 - Synthetic data was generated using a synthesizer based on the same source-filter decomposition that the proposed approach is designed to identify and recover.  The learned bases $U_i$ capture the space of variation in the synthetic data, but it's unclear how representative this is of real speech spectra.
  - Overall it is unclear how well the approach might generalize to real data which does not perfectly match the synthetic labeled datasets?  E.g., speech in the presence of background noise or overlapping speakers.

 - The fact that a VAE trained on clean speech spectrogram frames naturally disentangles this structure is of moderate interest to practitioners in that domain.
   But it's also not very surprising, as the problem of decomposing speech into separate source and filter components has been well understood for decades.
   See e.g., slides 15-18 of https://www.ee.columbia.edu/~dpwe/e6820/lectures/L05-speechmodels.pdf and references in https://en.wikipedia.org/wiki/Mel-frequency_cepstrum#History which describe how the cosine transform commonly used in cepstral analysis approximates the principal components of log-spectrum.
   Qualitatively, the described transformations in Sec 4 (transforming to constant f0, or "whispered" speech) are all fairly straightforward to implement without any machine learning at all using signal models from the 1980s, e.g., LPC or cepstral analysis, by modifying the source components while keeping the filter components fixed.
   Training a VAE and using a complex generative model such as WaveGlow for time-domain audio reconstruction is overkill for these simple transformations.

Specific comments:
- Sec 1, top of page 2: There have been other recent approaches to speech synthesis based on the source-filter model which are worth referencing in addition to LPCNet.  For example https://ieeexplore.ieee.org/abstract/document/8915761.
- Sec 4:  It's interesting that $M_0$  is as high as four, when the fundamental frequency should be easily captured by a scalar.  Similar for $M_2$ and $M_3$.  Is there an explanation for this?
- Sec 4.2.2: The low average NISQA scores of 2.6 (out of 5) on the original speech signals suggests a potential mismatch between the data that model was trained on (presumably real speech?), and the isolated vowel sounds used for evaluation.  Overall the large confidence intervals for NISQA in Table 1 further call the utility of this metric for this task into question.
- Sec 5, top of page 9: The papers cited as "voice conversion methods" all describe text-to-speech synthesis techniques, not approaches to pure text-independent voice conversion, an area which has been well studied for many years.  Some recent deep-learning based examples include the sequence-to-sequence-based models (e.g., https://arxiv.org/abs/1810.0686), or the CycleGAN-VC (http://www.kecl.ntt.co.jp/people/kameoka.hirokazu/publications/Kaneko2018EUSIPCO08_published.pdf) or StarGAN-VC (https://arxiv.org/abs/1907.12279) model families

**Summary Of The Paper:**

The paper proposes a method for utilizing labeled synthetic data in order to characterize and control the latent space of a VAE trained on individual frames of speech spectrograms.
Key properties of the data which one might want explicit control over are identified, i.e., pitch and formant frequencies, and a parametric speech synthesizer is used to generate synthetic datasets for each property, where the property in question is varied but all others are kept fixed.
These labeled data are used to identify subspaces of a VAE latent which correspond to each property, essentially by a principal components analysis of the latent vectors from each point in the synthetic dataset for that property.
The degree of disentanglement of the latent representation can be characterized by how orthogonal the bases are across subspaces.
Furthermore, individual properties can be directly controlled in isolation by learning a simple linear regression model mapping from the quantity in question (e.g., fundamental frequency in Hertz) to the subspace basis using supervision from the corresponding synthetic data.

**Summary Of The Review:**

Although the paper is well written, it's main findings seem likely to be of interest to only a small segment of the community.  Unless the proposed technique is more broadly applicable to more complicated domains, it seems like this work might be a better fit for a more specialized speech venue.

---

> ### Author Response · Authors · 2021-11-18
> **Response to Reviewer kWcG (Specific comments)**
>
> > **Sec 1, top of page 2: There have been other recent approaches to speech synthesis based on the source-filter model which are worth referencing in addition to LPCNet. For example https://ieeexplore.ieee.org/abstract/document/8915761.**
>
> We thank the Reviewer for this suggestion. We added the suggested reference and additional ones in the revised manuscript.
>
> > **Sec 4: It's interesting that M0 is as high as four, when the fundamental frequency should be easily captured by a scalar. Similar for M2 and M3. Is there an explanation for this?**
>
> We thank the Reviewer for this very relevant question. The latent manifolds encoding the fundamental and formant frequencies are actually nonlinear. As we approximate them using linear subspaces, we need slightly more dimensions than the intrinsic dimension of the nonlinear manifold (e.g., you can think of approximating a 1D spiral manifold by the 2D Euclidean space that contains it). This is not optimal in the sense that the latent representations of the fundamental and formant frequencies are slightly overparametrized, but it still allows us to learn accurate regression models for disentangled pitch and formant transformations. Developing invertible nonlinear manifold learning techniques is nevertheless an interesting perspective.
>
> > **Sec 4.2.2: The low average NISQA scores of 2.6 (out of 5) on the original speech signals suggests a potential mismatch between the data that model was trained on (presumably real speech?), and the isolated vowel sounds used for evaluation. Overall the large confidence intervals for NISQA in Table 1 further call the utility of this metric for this task into question.**
>
> We confirm that the VAE model was trained on real speech data, using the Wall Street Journal dataset. Evaluating the quality of generated/transformed data with deep generative models is intrinsically difficult due to the lack of ground truth, which is true independently of the application domain. We agree that NISQA is not perfect, but informal listening tests have confirmed that it correlates reasonably well with the subjective quality of the transformed speech signals evaluated in this paper.
>
>
> > Sec 5, top of page 9: The papers cited as "voice conversion methods" all describe text-to-speech synthesis techniques, not approaches to pure text-independent voice conversion, an area which has been well studied for many years. Some recent deep-learning based examples include the sequence-to-sequence-based models (e.g., https://arxiv.org/abs/1810.0686), or the CycleGAN-VC (http://www.kecl.ntt.co.jp/people/kameoka.hirokazu/publications/Kaneko2018EUSIPCO08_published.pdf) or StarGAN-VC (https://arxiv.org/abs/1907.12279) model families
>
> We thank the Reviewer for this suggestion. We added these references in the revised manuscript.
>
>
> > **Although the paper is well written, it's main findings seem likely to be of interest to only a small segment of the community.**
>
> As already mentioned and supported by the list of relevant topics in the call for papers, we believe that this submission is well within the scope of ICLR. With all due respect, we would also like to quote the Reviewer guide (https://iclr.cc/Conferences/2022/ReviewerGuide):
> “Don’t reject a paper just because you don’t find it “interesting”. This should not be a criterion at all for accepting/rejecting a paper. The research community is so big that somebody will find some value in the paper (maybe even a few years down the road), even if you don’t see it right now.”

---

> ### Author Response · Authors · 2021-11-18
> **Response to Reviewer kWcG (Part 4)**
>
>
> > **The fact that a VAE trained on clean speech spectrogram frames naturally disentangles this structure is of moderate interest to practitioners in that domain.**
>
> Interpretation and control of learned representations with deep generative models (especially VAEs and GANs) is currently an important topic in the machine learning community, regardless of the application domain, as confirmed by the list of relevant topics at ICLR. Previous works (see Related Work section of the submission) have mainly focused on natural images, very few addressed speech. However, there is currently a high interest in the use of VAEs for speech modeling and synthesis. Showing that a VAE naturally disentangles the source-filter structure offers several new possibilities for the practitioner, for instance to control the melodic aspect of the generated speech, or to inform speech enhancement, separation or more generally restoration methods using VAE-based speech priors.
>
>
> > **But it's also not very surprising, as the problem of decomposing speech into separate source and filter components has been well understood for decades.**
>
> We respectfully disagree with the reviewer, the fact that the source-filter decomposition of speech has been well understood for decades does not make the fact that a VAE learns this decomposition in a pure data-driven unsupervised approach “not very surprising”. We do think that it is a surprising result, especially considering that many papers (if not all) on disentangled image representation learning with VAEs (see Related Work section) have consistently reported that a vanilla VAE cannot learn disentangled representations without modifications in the loss function, for instance. Our approach in this study was to exploit well-understood signal processing models to make sense of the learned representation of a VAE. We believe that relying on more traditional and well-understood methods to improve the understanding and interpretation of recent deep learning models is an interesting approach. The results presented in this paper are also “surprising” in the sense that a single methodology works well for transforming both the fundamental and formant frequencies, while traditional signal processing methods use very different techniques to transform these attributes (e.g., manipulating the spectral envelope is very different from manipulating the excitation signal in LPC).
>
>
> > **See e.g., slides 15-18 of https://www.ee.columbia.edu/~dpwe/e6820/lectures/L05-speechmodels.pdf and references in https://en.wikipedia.org/wiki/Mel-frequency_cepstrum#History which describe how the cosine transform commonly used in cepstral analysis approximates the principal components of log-spectrum.**
>
> We are not sure to understand the connection between this remark and the proposed approach. The DCT in cepstral analysis indeed allows for compressing the log-spectrum in a few coefficients that are relatively uncorrelated. But it does not allow one to easily control the shape of the speech log-spectra (regarding pitch and formant attributes) in an analysis-transformation-synthesis framework.
>
>
> > **Qualitatively, the described transformations in Sec 4 (transforming to constant f0, or "whispered" speech) are all fairly straightforward to implement without any machine learning at all using signal models from the 1980s, e.g., LPC or cepstral analysis, by modifying the source components while keeping the filter components fixed. Training a VAE and using a complex generative model such as WaveGlow for time-domain audio reconstruction is overkill for these simple transformations.**
>
> We totally agree with the reviewer, and as mentioned in the discussion of the experimental results, the objective of this paper is not to compete with traditional signal processing methods. It is rather to advance on the understanding of deep generative modeling of speech signals and to compare honestly with highly-specialized traditional signal processing systems. As already mentioned, we believe that this objective is well aligned with the scope of ICLR.
> Speech transformation methods originally based on signal processing techniques are currently largely revisited with deep learning techniques, so we believe that this submission is totally relevant and timely in this context. Moreover, most deep-learning-based methods indeed focus on more complex speech transformations but “without opening the black box”. The present paper advances on the interpretability of the models, applied on simple transformations that could however open the road to more advanced ones, e.g. prosodic transformations.

---

> > ### Comment · Reviewer_kWcG · 2021-11-30
> > **Response to rebuttal**
> >
> > >> The fact that a VAE trained on clean speech spectrogram frames naturally disentangles this structure is of moderate interest to practitioners in that domain.
> > >
> > > Interpretation and control of learned representations with deep generative models (especially VAEs and GANs) is currently an important topic in the machine learning community, regardless of the application domain, as confirmed by the list of relevant topics at ICLR. Previous works (see Related Work section of the submission) have mainly focused on natural images, very few addressed speech. However, there is currently a high interest in the use of VAEs for speech modeling and synthesis. Showing that a VAE naturally disentangles the source-filter structure offers several new possibilities for the practitioner, for instance to control the melodic aspect of the generated speech, or to inform speech enhancement, separation or more generally restoration methods using VAE-based speech priors.
> >
> > Just to clarify my position on the limitations of the particular domain addressed in this paper, which was perhaps not totally clear: it addresses the domain of *isolated spectrogram frames* of voiced speech, i.e., 64 ms segments of audio within vowels).   The analogous problem in image generation would be to model (components of?) isolated objects without surrounding context.
> >
> > I certainly agree that speech modeling should be and is within the purview of ICLR.  My concern was that the empirical results in this paper relate to a relatively narrow and (I believe) well understood subset of the speech generation problem.
> >
> > That said, I certainly take the author's point that similar work regarding disentanglement of relatively simple problems in the image domain, e.g. on the dSprites dataset, are also of interest to the ICLR community.
> >
> > The fact that the proposed method can also be applied to that domain (as described in the [comment above](https://openreview.net/forum?id=zxEfpcmTDnF&noteId=NlKhpX5h5At) alleviates many of my concerns about the limited applicability of the proposed approach.  For whatever it's worth, I think that the paper would be *much* stronger if presented itself as a general approach to weakly-supervised disentanglement and included  empirical results across both voiced speech spectrogram frames and dSprites or similar images.
> >
> > >> See e.g., slides 15-18 of https://www.ee.columbia.edu/~dpwe/e6820/lectures/L05-speechmodels.pdf and references in https://en.wikipedia.org/wiki/Mel-frequency_cepstrum#History which describe how the cosine transform commonly used in cepstral analysis approximates the principal components of log-spectrum.
> > >
> > > We are not sure to understand the connection between this remark and the proposed approach. The DCT in cepstral analysis indeed allows for compressing the log-spectrum in a few coefficients that are relatively uncorrelated. But it does not allow one to easily control the shape of the speech log-spectra (regarding pitch and formant attributes) in an analysis-transformation-synthesis framework.
> >
> > One can indeed easily control the f_0, independently of the formants, within a speech frame (at least in terms of spectrogram magnitude) from the cepstrum, by simply shifting the pitch pulse component (corresponding more or less to an isolated impulse in high "quefrency", see the bottom figure of slide 16 of https://www.ee.columbia.edu/~dpwe/e6820/lectures/L05-speechmodels.pdf) along the quefrency axis.  I agree that there is no equally trivial transformation of the low quefrency components that isolate individual formants.  However it should be possible to use an approach proposed in Section 3.3 of this paper to learn the latent subspace which corresponds to each formant.
> >
> > Concretely, my guess is that replacing the learned VAE encoder and decoder in Fig 4. with the real cepstrum and its inverse (essentially a shallow 2 layer network with a single log(abs(.)) nonlinearity) will result in disentanglement performance that is comparable to the learned VAE.  I can't claim to have run this experiment, so I may be wrong.  But it would be quite interesting to see if indeed the VAE is learning something better than this.  Then again the paper's novelty lies in the technique for identifying subspaces of the latent subspace which correspond to properties of interest, not in the VAE itself.

---

> ### Author Response · Authors · 2021-11-18
> **Response to Reviewer kWcG (Part 3)**
>
> Regarding more complex speech tasks, the proposed generative model could be used as a speech prior for pitch-informed speech separation or denoising. Several methods have recently investigated the use of VAE-based speech priors for speech separation and denoising. However, the lack of interpretability of the VAE latent space prevents the use of external information to guide the separation/denoising process. For example, in the speech separation method of Kameoka et al. (2019), the conditional prior in Equation (9) of the submitted manuscript could be used to constrain the optimization of the VAE latent variable, given input pitch contours for the different speakers (assuming that multi-speaker pitch estimation can be used as a preprocessing step to multi-speaker speech separation). With a VAE trained on multi-microphone and reverberant speech data, the proposed approach could also be extended to control acoustic parameters such as the reverberation time or the source direction of arrival. More generally, we believe that the proposed method will be helpful to develop more expressive and interpretable deep generative speech priors applied to different inverse problems in speech processing. This aspect was already mentioned in the introduction of the submitted paper. We added a reference to Kameoka et al. (2019).
>
> > **Synthetic data was generated using a synthesizer based on the same source-filter decomposition that the proposed approach is designed to identify and recover.**
>
> We respectfully disagree with the Reviewer on this point, as the proposed method relies on a VAE that was trained without any supervision and on real speech data. It was absolutely not guaranteed a priori that this VAE, trained in an unsupervised fashion, would organize its latent space so as to encode the fundamental and formant frequencies in orthogonal subspaces. We believe that this is an interesting finding regarding the interpretation of VAE learned representations, as it shows that a vanilla VAE, without any supervision, learns a latent representation that is consistent with the source-filter model of human speech production. To the best of our knowledge, this was never shown before in the literature.
>
> > **The learned bases Ui capture the space of variation in the synthetic data, but it's unclear how representative this is of real speech spectra. Overall it is unclear how well the approach might generalize to real data which does not perfectly match the synthetic labeled datasets? E.g., speech in the presence of background noise or overlapping speakers.**
>
> The learned bases Ui indeed capture the space of variation in the synthetic data, but once projected in the latent space of the VAE trained on real speech data. Moreover, the experimental Section 4.2.2 precisely shows that the proposed method generalizes well to real speech spectra. It therefore validates that the latent subspaces learned from synthetic data are also representative of real speech spectra. Otherwise, we would not be able to accurately perform disentangled pitch and formant transformations on real speech spectra.
> We insist on the fact that we used only about 3-to-6 seconds of artificial speech data. We believe that using such little supervision in order to learn the latent subspaces and control the corresponding attributes in the latent space of an unsupervised VAE trained on a large-scale unlabeled speech dataset is a strength of the approach.
> Regarding the applicability of the speech transformation proposed method in the presence of background noise or overlapping speakers, as already mentioned above, the proposed deep generative speech model which is conditioned on the fundamental frequency could be used in recent speech enhancement and separation methods that exploit VAEs as speech priors, e.g. in Kameoka et al. (2019) to perform pitch-informed speech separation.

---

> ### Author Response · Authors · 2021-11-18
> **Response to Reviewer kWcG (Part 2)**
>
> > **At the very least the paper could use a detailed discussion about the how the proposed approach might be applied to other domains of interest to the generative modeling community, e.g., to natural images or at least to more complex speech tasks. Such generalizations seems difficult since a key requirement for the technique to work is the existence of datasets which capture variation in a single attribute independently of others (or a synthesizer/generator which can create such datasets), i.e., it requires data which explicitly demonstrates disentanglement to use as supervision. This requirement seems like it would be difficult to fulfill in other less restrictive domains.**
>
> Regarding other domains, the proposed approach might be applied to natural images, for instance to translate, rotate and scale objects in natural images, or to adjust brightness, similarly to applications in related works on controlled deep generative image modeling (Jahanian et al., 2019; Plumerault et al., 2020; Goetschalckx et al., 2019; Härkönen et al., 2020). On face images, it could be used to perform expressive transformations, such as making people smile. As pointed out by the Reviewer, a key requirement would be to be able to generate image examples that capture variations in a single attribute, independently of others. Contrary to what the Reviewer suggests, we believe that creating synthetic images is not a difficult task. We also want to emphasize that the proposed method, when applied to speech data, required only a very few synthetic examples to learn the speech attributes latent subspaces and corresponding regression models. In our experiments, we used trajectories of about 200 to 400 examples to learn the subspace and regression model of each speech attribute. This corresponds to between 3.2 and 6.4 seconds of controlled labeled speech data, to be compared with the 25 hours of unlabeled speech data used in the unsupervised training of the VAE (Wall Street Journal dataset). We could probably quite easily create a few hundred images with an object that is translated, rotated or scaled, so as to control the latent space of a VAE trained on a large-scale unlabeled natural image dataset. Due to space constraints, we have only briefly discussed applications of the proposed method to image data in the conclusion of the revised manuscript. We would also like to mention preliminary results we obtained on the dSprites dataset (https://github.com/deepmind/dsprites-dataset) that show that the proposed method can indeed be applied to precisely control position and scale of objects in images, given target values for these attributes to be modified (which is a key novelty compared with previous methods on disentangled representation learning with VAEs, see the “Related Work” section). These preliminary results are available at https://docs.google.com/presentation/d/e/2PACX-1vTkEIO4s1lm1nD7XO-fGffyZ7XMaFakyJjTewX1XYkCw67rfoRz4EizEA-jJM5FjH5n50NUGlN1GyBA/pub
> We nevertheless want to insist on the fact that the main focus of this paper is to analyze and control a VAE applied to the modeling of speech signals, which we believe is not less interesting than applications on images, and more interesting than reporting results on the dSprites dataset. In fact, the present submission matches with multiple items of the list of relevant topics indicated in the ICLR 2022 call for papers (https://iclr.cc/Conferences/2022/CallForPapers), namely “unsupervised, semi-supervised, and supervised representation learning”; “visualization or interpretation of learned representations”; “applications in audio, speech, robotics, neuroscience, computational biology, or any other field”.

---

> ### Author Response · Authors · 2021-11-18
> **Response to Reviewer kWcG (Part 1)**
>
>
>
> > **Experiments are all on a relatively simple domain of speech spectrogram frames (in fact, restricted only to vowels, speech sounds which are most easily modeled using a source-filter decomposition). The quantitative results (sec 4.2) are further restricted to isolated vowels, more closely matching the domain of the labeled synthetic data and not covering the dynamic space of variation in natural speech.**
>
> Evaluating quantitatively the quality of generated/transformed data with deep generative models is difficult, not only for speech data but also for images. We chose this dataset of English vowels because it is labeled with pitch and formant frequencies, which is required to evaluate the methods. Nevertheless, we performed additional experiments on speech utterances from TIMIT, so as to complexify the test data. A difficulty is that TIMIT utterances are not labeled with pitch and formant information, so we replaced the ground truth by estimates computed with CREPE for the pitch and PRAAT for the formant frequencies. The results (available in Appendix D.3 of the revised paper) are thus slightly less reliable than with the English vowels dataset. Yet, we observe that these new results are totally consistent with the ones on the English vowels, which strengthens the conclusions drawn in the experimental section of the paper.

---

### Official Review · Reviewer_pLaG · 2021-11-03

**Correctness:** 3
**Technical Novelty And Significance:** 3
**Empirical Novelty And Significance:** 3
**Recommendation:** 6
**Confidence:** 4

**Main Review:**

The paper is interesting in analyzing the subspaces of the latent space that correspond to each formant and the pitch in a trained VAE. The subspaces were obtained using synthesized speech which is not ideal and it is not clear whether the method would work well on all speech data. Another possible issue is that the model performance is analyzed with very simplistic data: English vowels. It was not tried on continuous real speech utterances. Also, I am not totally convinced that this latent space post-analysis way is the best way to obtain disentangled latent information. It could be possible to have multiple embeddings that correspond to each fj from the start. How would this approach compare to the post-analysis approach? The subspaces are found to be "mostly" orthogonal but not exactly, so there may be some leakage between subspaces.

Some itemized questions and recommendations:
1. Dimension D of x_i is not defined before it was mentioned in line 7 of Section 2.1. Maybe instead of {\cal X}, you can directly indicate the vector space for the features.
2. In Section 3.3, the justification for linear regression for predicting the subspace coefficients is not clear to me. Why would the subspace coefficients be a scalar multiple of the fj value? Would it not work better if you used a multi-layer nonlinear prediction?

**Summary Of The Paper:**

This paper analyzes VAE latent embeddings to extract subspaces that relate to pitch (f0) and formant frequencies (f1 through fN). This is done through first training a frame-synchronous  IS-VAE model from clean speech data. The authors pass controlled synthesized speech through the model and obtain the embeddings corresponding to the synthesized data with randomly varying one fj component and keeping others constant. Each embedding corresponds to a frame of spectral data. The set of embeddings are then analyzed by PCA to find the principal eigenvectors corresponding to each fj explaining 80% of the total energy. The mapping from f0, f1, .. fN values and the subspace coefficients is done through a linear regression mapping independently for each fj which is also learned from synthetic data. Through simple examples, the authors show that they can change individual components (f0, f1, .., fN) by modifying the embeddings correspond to a signal, to some extent. They compare their results with rule-based vocoders (WORLD, TD-PSOLA) and a VAE baseline that does not use subspaces.

**Summary Of The Review:**

The analysis looks interesting, but the training and eval data used for this analysis looks too simplistic. A direct disentangled representation could also be possible with an easy way to make smaller embeddings corresponding to each fj.

---

> ### Author Response · Authors · 2021-11-18
> **Response to Reviewer pLaG (Part 2)**
>
> > **Dimension D of x_i is not defined before it was mentioned in line 7 of Section 2.1. Maybe instead of {\cal X}, you can directly indicate the vector space for the features.**
>
> We used {\cal X} to remain general at the beginning of Section 2.1. Indeed, later on in this section (third paragraph) we write that the observed data vector x is non-negative, as it corresponds to a speech power spectrum, but generally a VAE is not restricted to non-negative data. Following your suggestion, to be more precise we changed the first sentence of Section 2.1 and wrote “$\mathbf{x} \in \mathcal{X} \subset \mathbb{R}^D$.
>
> > **In Section 3.3, the justification for linear regression for predicting the subspace coefficients is not clear to me. Why would the subspace coefficients be a scalar multiple of the fj value? Would it not work better if you used a multi-layer nonlinear prediction?**
>
> The regression model is actually piecewise linear (thus nonlinear), which is more expressive than a simple linear model. As rightly suggested by the Reviewer, a simple linear model would perform poorly here. Using a multi-layer neural network would probably be a suitable alternative, but we prefered to keep a piecewise linear regression model as it is simpler and works well in practice for this task.

---

> > ### Comment · Reviewer_pLaG · 2021-11-30
> > **Response to rebuttal**
> >
> > I read the authors' response and other reviews. Thanks for the responses to my review comments.
> >
> > The issue about having separate embeddings for each fj is a valid approach if the aim of the paper is to develop a system that allows f0 and formant modification. However the authors claim to have a scientific curiosity to show that trained VAE embeddings contain that information already and it may be important to post-disentangle this information from these embeddings. My comment was mostly referring to the fact that if the goal is to build a system for speech modification, this may not be the best way. If this is true, then the paper's main contribution becomes resolving some kind of scientific curiosity and it is debatable how valuable this is from an application point of view. The fact that VAE embeddings already contain some disentangle-able information about f0 and formants seems scientifically interesting to show but it is not difficult to guess that that could be the case beforehand as well.

---

> ### Author Response · Authors · 2021-11-18
> **Response to Reviewer pLaG (Part 1)**
>
> > **The subspaces were obtained using synthesized speech which is not ideal and it is not clear whether the method would work well on all speech data.**
>
> The subspaces are indeed obtained using synthesized speech data, but the VAE was originally trained on real speech data (Wall Street Journal dataset), so the proposed method applies to, and is evaluated on, real and not synthesized speech signals. The artificial speech data are only used as a weak source of supervision to learn the subspaces and the regression models from the unsupervised VAE trained on real speech signals.
> We would like to emphasize that we used only about 3-to-6 seconds of synthetic labeled speech data, to be compared with the 25 hours of real unlabeled speech data used to train the unsupervised VAE. We believe that using such little supervision in order to learn the latent subspaces and control the corresponding speech attributes in the latent space of the VAE is a strength of the approach.
>
> > **Another possible issue is that the model performance is analyzed with very simplistic data: English vowels. It was not tried on continuous real speech utterances.**
>
> Evaluating quantitatively the quality of generated/transformed data with deep generative models is difficult, not only for speech data but also for images. We chose this dataset of English vowels because it is labeled with pitch and formant frequencies, which is required to evaluate the methods. Nevertheless, we performed additional experiments on speech utterances from TIMIT, so as to complexify the test data. A difficulty is that TIMIT utterances are not labeled with pitch and formant information, so we replaced the ground truth by estimates computed with CREPE for the pitch and PRAAT for the formant frequencies. The results (available in Appendix D.3 of the revised paper) are thus slightly less reliable than with the English vowels dataset. Yet, we observe that these new results are completely consistent with the ones on the English vowels, which strengthens the conclusions drawn in the experimental section of the paper.
> Please also note that qualitative results include examples on continuous real speech utterances. The Reviewer is kindly referred to Figure 2.c, Figure 8, and the companion website (https://tinyurl.com/iclr2022).
>
> > **Also, I am not totally convinced that this latent space post-analysis way is the best way to obtain disentangled latent information. It could be possible to have multiple embeddings that correspond to each fj from the start. How would this approach compare to the post-analysis approach? The subspaces are found to be "mostly" orthogonal but not exactly, so there may be some leakage between subspaces.**
>
> We are not sure to understand what the Reviewer means by “It could be possible to have multiple embeddings that correspond to each fj from the start. How would this approach compare to the post-analysis approach?.” We believe that the proposed “post-analysis” approach is a strength of the paper, because it shows that a vanilla VAE naturally encodes the fundamental frequency and formant frequencies in orthogonal latent subspaces, which is coherent with the human voice production system. This structure of the VAE latent space naturally emerges in a bottom-up fashion, which we believe was not obvious, not reported in the existing literature, and is therefore interesting to report. “Pre-enforcing” the disentanglement in the VAE latent space would require modifying the VAE model and/or the objective function using expert knowledge about the structure of speech signals, in a top-down approach. As suggested by the Reviewer, this may lead to a latent representation even more disentangled, maybe with perfectly or closer to perfectly orthogonal subspaces, but one of the main contribution of this paper is precisely to show that this disentangled structure of the latent space naturally emerges when the VAE is trained on real speech signals without any supervision or specific disentanglement-enforcing constraint. We believe this is an important step towards a better understanding of deep generative modeling of speech.

---

### Author Response · Authors · 2021-11-18
**Updates**

We thank the Reviewers for their valuable feedback and the time they dedicated to review our submission. You will find below the response to the concerns raised by each individual Reviewer. The main changes in the revised paper are the following:

- We added an appendix with additional experiments on the TIMIT dataset;
- We briefly discuss in the conclusion how the proposed method can be applied to images;
- We highlighted the small quantity of labeled data used in the proposed method (a few seconds of artificially generated labeled speech, to be compared with the 25 hours of unlabeled data used to train the VAE in an unsupervised manner);
- We added the references suggested by the reviewers.

Reviewers kWcG, e7Cb and vTAw all raised the concern that this paper might be of limited interest to the general audience of ICLR. However, the present submission matches with multiple items of the list of relevant topics indicated in the ICLR 2022 call for papers, namely “unsupervised, semi-supervised, and supervised representation learning;” “visualization or interpretation of learned representations;” “applications in audio, speech, robotics, neuroscience, computational biology, or any other field.” Interpretation and control of learned representations with deep generative models is currently an important topic in the machine learning community, regardless of the application domain. Previous works (see Related Work section of the submission) have mainly focused on natural images, and most of them were published in machine learning and not computer vision conferences. In this work, we propose a new methodology to analyze and control the latent space of a VAE, which we apply to the modeling of speech signals. We believe this is as interesting as any application on images.

---

### Decision · Program_Chairs · 2022-01-20

**Decision:**

Reject

**Comment:**

This work shows that the source-filter model of speech production naturally arises in the latent space of a variational autoencoder (VAE). It is interesting that the fundamental frequency and formant frequencies are encoded in orthogonal subspaces of the VAE latent space -- this opens up a possible way of easily controlling these.

The key motivation/goal of the paper has caused some confusion. The abstract highlights an observation about VAE’s learned representation. In retrospection, some reviewers have not found the findings very surprising. On the other hand, the authors also do not attempt at developing and evaluating a speech generation method. As is, the paper seems to be much more suitable to a specialized workshop on speech. Alternatively, the paper could be extended to other modalities to show steerability of a representation using a synthetic dataset. However, the current scope seems to be somewhat limited hence I am not able to recommend the current manuscript for acceptance.